# Fine-grained Optimization of Deep Neural Networks

**Mete Ozay**[*]

## Abstract

In recent studies, several asymptotic upper bounds on generalization errors on deep neural networks (DNNs) are theoretically derived. These bounds are functions of several norms of weights of the DNNs, such as the Frobenius and spectral norms, and they are computed for weights grouped according to either input and output channels of the DNNs. In this work, we conjecture that if we can impose multiple constraints on weights of DNNs to upper bound the norms of the weights, and train the DNNs with these weights, then we can attain empirical generalization errors closer to the derived theoretical bounds, and improve accuracy of the DNNs.

To this end, we pose two problems. First, we aim to obtain weights whose different norms are all upper bounded by a constant number. To achieve these bounds, we propose a two-stage renormalization procedure; (i) normalization of weights according to different norms used in the bounds, and (ii) reparameterization of the normalized weights to set a constant and finite upper bound of their norms. In the second problem, we consider training DNNs with these renormalized weights. To this end, we first propose a strategy to construct joint spaces (manifolds) of weights according to different constraints in DNNs. Next, we propose a fine-grained SGD algorithm (FG-SGD) for optimization on the weight manifolds to train DNNs with assurance of convergence to minima. Experimental analyses show that image classification accuracy of baseline DNNs can be boosted using FG-SGD on collections of manifolds identified by multiple constraints.

## 1 Introduction

Understanding generalization behavior of DNNs is an open problem [1]. Recent works [2–8] addressed this problem by extending the early results proposed for shallow linear neural networks (NNs) [9] for a more general class of DNNs (e.g. neural networks with ReLU), recurrent neural networks [10, 11], and convolutional neural networks (CNNs) (see Table 1 for a comparison). The proposed asymptotic bounds were obtained by defining weight matrices of DNNs using random matrices, and applying concentration inequalities on them. Thereby, the bounds were computed by functions of several $\ell_p$ norms of these matrices, for $1 \le p \le \infty$.

In this work, we conjecture that if we can impose multiple constraints on weights of DNNs to set upper bounds of the norms of the weight matrices, and train the DNNs with these weights, then the DNNs can achieve empirical generalization errors closer to the proposed theoretical bounds, and we can improve their accuracy in various tasks. We pose two problems in order to achieve this goal;

1. Renormalization of weights to upper bound norms of their matrices.
2. Training DNNs with renormalized weights with assurance to convergence to minima.

### 1.1 Background

Spaces of normalized weights can be identified by different Riemann manifolds [12]; (i) unit norm weights reside on the sphere $Sp(A_l B_l - 1)$, (ii) orthonormal weights belong to the Stiefel manifold

---

[*]meteozay@gmail.com

$St(A_l, B_l)$, and (iii) weights with orthogonal columns reside on the oblique manifold $Ob(A_l B_l)$, at each $l^{th}$ layer of a DNN (formal mathematical definitions are given in the supplemental material).

**Challenge of training DNNs with multiple constraints:** DNNs can be trained with multiple constraints using optimization methods proposed for training shallow algorithms [13, 14], and individual manifolds [12, 15]. If we employ these methods on products of weight manifolds (POMs) to train DNNs, then we observe early divergence, vanishing and exploding gradients due to nonlinear geometry of product of different manifolds.

More precisely, the assumption of a bound on the operator norm of Hessian of geodesics in POMs, which is required for assurance of convergence, fails, while performing Stochastic Gradient Descent (SGD) with backpropagation on product of different weight manifolds. Therefore, a non-increasing bound on the probability of failure of the optimization algorithm cannot be computed, and a convergence bound cannot be obtained.

We consider training DNNs using a more general setting employing groups of weights which can be normalized according to different normalization constraints. Group wise operations are implemented by concatenating weight matrices $\omega_{g,l}^i$ belonging to each $g^{th}$ group by $\omega_{g,l} = (\omega_{g,l}^1, \omega_{g,l}^2, \ldots, \omega_{g,l}^{\mathfrak{g}}), \forall g = 1, 2, \ldots, G_l$. For the corresponding group, a space of concatenated weights is identified by Cartesian product of manifolds of weights $\omega_{g,l}^i, i = 1, 2, \ldots, \mathfrak{g}$. In addition, if we renormalize weights using standard deviation of features obtained at each epoch, then geometry of the manifolds of weights also changes. Therefore, we address the second subproblem which is optimization on dynamically changing product manifolds of renormalized weights.

In order to solve these problems, we first propose a mathematical framework to make use of the geometric relationship between weight manifolds determined by different constraints (Section 5). Then, we suggest an approach for training DNNs using multiple constraints on weights to improve their performance under the proposed framework. To this end, we propose a new algorithm that we call fine-grained stochastic gradient descent (FG-SGD) to train DNNs using POMs. We elucidate geometric properties of POMs to assure convergence of FG-SGD to global minima while training nonlinear DNNs with particular assumptions on their architectures, and to local minima while training a more generic class of nonlinear DNNs.

## 2 A Conceptual Overview of the Contributions

**(1) Bounding norms of weights:** We propose a two-stage renormalization procedure. First, we normalize weights according to the Euclidean, Frobenius and spectral norm, since they are used in the bounds of generalization errors [2–8]. Second, we aim to reparameterize the normalized weights to set a finite and constant upper bound on the weight matrices. For this purpose, we can use a parameter learning approach as utilized in batch normalization (BN) [16]. However, such an approach substantially increases running time of DNNs during training. In addition, it is not efficient to estimate the parameters using small number of samples in batch training. Therefore, we reparameterize weights according to (a) geometric properties of weight spaces, and (b) statistical properties of features (standard deviation) on which the weights are applied. Using (b), we can also decrease computation time of statistical properties especially when we apply separable operations such as channel/depth wise separable convolutions [17–19].

The proposed reparameterization method enables to set upper bound of each different norm of weight matrices to 1.0. In addition, the proposed renormalization procedure enables to control variance of weights during training of DNNs, thereby assures that DNNs do not have spurious local minima [20]. Employment of standard deviation in reparameterization also makes optimization landscapes significantly smoother by bounding amount of change of norms of gradients during training. This property has been recently studied to analyze effect of BN on optimization landscape in [21]. We use this property to develop a new optimization method for weight renormalization in this paper, as explained in the next problem.

**(2) Training DNNs with renormalized weights:** Note that, there is not a single procedure used to normalize weights jointly according to all different norms. Thereby, we normalize weights in groups such that similar or different norms can be used to normalize matrices of weights belonging to each different group.

We can mathematically prove that the procedure proposed to solve the previous problem (1) can set an upper bound for all of the aforementioned norms. However, we do not have a mathematical proof to explain whether weights normalized according a single norm can provide the best generalization bound, and to determine its type. We examine this question in various experiments in detail in the supp. mat. Since we cannot mathematically verify this observation, we conjecture that using a diverse set of weights normalized with different constraints improves the generalization error compared to using weights normalized according to single constraint. We consider mathematical characterization of this property as an open problem.

Our contributions are summarized as follows:

1. DNNs trained using weights renormalized by the proposed method can achieve tighter bounds for theoretical generalization errors compared to using unnormalized weights (see Proposition 1 in the supp. mat. for derivation). These DNNs do not have spurious local minima [20] (see the next section for a detailed discussion). The proposed scaling method generalizes the scaling method proposed in [22] for weight normalization by incorporating geometric properties of weight manifolds.

2. We explicate the geometry of weight manifolds defined by multiple constraints in DNNs. For this purpose, we explore the relationship between geometric properties of POMs (i.e. sectional curvature), gradients computed at POMs (Theorem 1), and those of component manifolds of weights in DNNs in Section 5 (please see Lemma 1 in the supp. mat. for more precise results).

3. We propose an algorithm (FG-SGD) for optimization on different collections of POMs (Section 5) by generalizing SGD methods employed on weight manifolds [12, 23]. Next, we explore the effect of geometric properties of the POMs on the convergence of the FG-SGD using our theoretical results. In the proof of convergence theorems, we observe that gradients of weights should satisfy a particular normalization requirement and we employ this requirement for adaptive computation of step size of the FG-SGD (see (5) in Section 5.2.2). We also provide an example for computation of a step size function for optimization on POMs identified by the sphere (Corollary 2 in the supp. mat.).

4. We prove that loss functions of DNNs trained using the proposed FG-SGD converges to minima almost surely (see Theorem 2 and Corollary 1 in the supplemental material).

## 3 Construction of Sets of POMs in DNNs

Let $S = \{s_i = (\mathbf{I}_i, y_i)\}_{i=1}^N$ be a set of training samples, where $y_i$ is a class label of the $i^{th}$ image $\mathbf{I}_i$. We consider an $L$-layer DNN consisting of a set of tensors $\mathcal{W} = \{\mathcal{W}_l\}_{l=1}^L$, where $\mathcal{W}_l = \{\mathbf{W}_{d,l} \in \mathbb{R}^{A_l \times B_l \times C_l}\}_{d=1}^{D_l}$, and $\mathbf{W}_{d,l} = [W_{c,d,l} \in \mathbb{R}^{A_l \times B_l}]_{c=1}^{C_l}$ is a tensor[2] of weight matrices $W_{c,d,l}, \forall l = 1, 2, \ldots, L$, for each $c^{th}$ channel $c = 1, 2, \ldots, C_l$ and each $d^{th}$ weight $d = 1, 2, \ldots, D_l$. In popular DNNs, weights with $A_l = 1$ and $B_l = 1$ are used at fully connected layers, and those with $A_l > 1$ or $B_l > 1$ are used at convolutional layers. At each $l^{th}$ layer, a feature representation $f_l(\mathbf{X}_l; \mathcal{W}_l)$ is computed by compositionally employing non-linear functions by

$$f_l(\mathbf{X}_l; \mathcal{W}_l) = f_l(\cdot; \mathcal{W}_l) \circ f_{l-1}(\cdot; \mathcal{W}_{l-1}) \circ \cdots \circ f_1(\mathbf{X}_1; \mathcal{W}_1), \tag{1}$$

where $\mathbf{X}_l = [X_{c,l}]_{c=1}^{C_l}$, and $\mathbf{X}_1 := \mathbf{I}$ is an image at the first layer ($l = 1$). The $c^{th}$ channel of the data matrix $X_{c,l}$ is convolved with the kernel $W_{c,d,l}$ to obtain the $d^{th}$ feature map $X_{c,l+1} := q(\hat{X}_{d,l})$ by $\hat{X}_{d,l} = W_{c,d,l} * X_{c,l}, \forall c, d, l$, where $q(\cdot)$ is a non-linear function, such as ReLU.

Previous works [12, 23] employ SGD using weights each of which reside on a single manifold[3] at each layer of a DNN. We extend this approach considering that each weight can reside on an individual manifold or on collections of products of manifolds, which are defined next.

**Definition 1** (Products of weight manifolds and their collections)**.** Suppose that $\mathcal{G}_l = \{\mathcal{M}_{\iota,l} : \iota \in \mathcal{I}_{\mathcal{G}_l}\}$ is a set of weight manifolds[3] $\mathcal{M}_{\iota,l}$ of dimension $n_{\iota,l}$, which is identified by a set of indices $\mathcal{I}_{\mathcal{G}_l}, \forall l = 1, 2, \ldots, L$. More concretely, $\mathcal{I}_{\mathcal{G}_l}$ contains indices each of which represents an identity number ($\iota$) of a weight that resides on a manifold $\mathcal{M}_{\iota,l}$ at the $l^{th}$ layer. In addition, a subset $\mathcal{I}_l^g \subseteq \mathcal{I}_{\mathcal{G}_l}, g = 1, 2, \ldots, G_l$, is used to determine a subset $\mathcal{G}_l^g \subseteq \mathcal{G}_l$ of weight manifolds which will be

Table 1: Comparison of generalization bounds. $\mathcal{O}$ denotes big-O and $\tilde{\mathcal{O}}$ is soft-O. $\delta_{l,F}$, $\delta_{l,2}$, and $\delta_{l,2\to1}$ denotes upper bounds of the Frobenius norm $\|\omega_l\|_F \le \delta_{l,F}$, spectral norm $\|\omega_l\|_2 \le \delta_{l,2}$ and the sum of the Euclidean norms for all rows $\|\omega_l\|_{2\to1} \le \delta_{l,2\to1}$ ($\ell_{2\to1}$) of weights $\omega_l$ at the $l^{th}$ layer of an $L$ layer DNN using $N$ samples.

| | **DNNs** (dynamic group scaling) |
|---|---|
| Neyshabur et al. [24] | $\mathcal{O}\left(\dfrac{2^L \prod\limits_{l=1}^{L} \prod\limits_{g=1}^{G_l} \delta_{g,l,F}}{\sqrt{N}}\right)$ |
| Bartlett et al. [3] | $\tilde{\mathcal{O}}\left(\dfrac{\prod\limits_{l=1}^{L} \prod\limits_{g=1}^{G_l} \delta_{g,l,2}}{\sqrt{N}}\left(\sum\limits_{l=1}^{L}\prod\limits_{g=1}^{G_l}\left(\dfrac{\delta_{g,l,2\to1}}{\delta_{g,l,2}}\right)^{\frac{2}{3}}\right)^{\frac{3}{2}}\right)$ |
| Neyshabur et al. [8] | $\tilde{\mathcal{O}}\left(\dfrac{\prod\limits_{l=1}^{L}\prod\limits_{g=1}^{G_l}\delta_{g,l,2}}{\sqrt{N}}\sqrt{L^2\varpi \sum\limits_{l=1}^{L}\prod\limits_{g=1}^{G_l}\dfrac{\delta_{g,l,F}^2}{\delta_{g,l,2}^2}}\right)$ |

Table 2: Comparison of norms of weights belonging to different weight manifolds. Suppose that weights $\omega_{g,l}^i \in \mathbb{R}^{A_l \times B_l}$ belonging to the $g^{th}$ group of size $|\mathfrak{g}|$, $g = 1, 2, \ldots, G_l$, $\forall l$ have the same size $A_l \times B_l$ for simplicity, and $\sigma(\omega_{g,l}^i)$ denotes the top singular value of $\omega_{g,l}^i$. $\|\omega_{g,l}^i\|_F$, $\|\omega_{g,l}^i\|_2$, and $\|\omega_{g,l}^i\|_{2\to1}$, denotes respectively the Frobenius, spectral and $\ell_{2\to1}$ norms of the weight $\omega_{g,l}^i$.

| Norms | (i) Sphere | (ii) Stiefel | (iii) Oblique |
|---|---|---|---|
| $\|\omega_{g,l}^i\|_2$ | $\sigma(\omega_{g,l}^i)$ | 1.0 | $\sigma(\omega_{g,l}^i)$ |
| $\|\omega_{g,l}^i\|_F$ | 1.0 | $(B_l)^{1/2}$ | $(B_l)^{1/2}$ |
| $\|\omega_{g,l}^i\|_{2\to1}$ | 1.0 | $(B_l)^{1/4}$ | $(B_l)^{1/4}$ |

aggregated to construct a product of weight manifolds (POM). Each $\mathcal{M}_{\iota,l} \in \mathcal{G}_l^g$ is called a component manifold of a product of weight manifolds which is denoted by $\mathbb{M}_{g,l}$. A weight $\omega_{g,l} \in \mathbb{M}_{g,l}$ is obtained by concatenating weights belonging to $\mathcal{M}_{\iota,l}$, $\forall \iota \in \mathcal{I}_l^g$, using $\omega_{g,l} = (\omega_1, \omega_2, \cdots, \omega_{|\mathcal{I}_l^g|})$, where $|\mathcal{I}_l^g|$ is the cardinality of $\mathcal{I}_l^g$. A $\mathcal{G}_l$ is called a *collection of POMs*. ∎

We propose three schemes called POMs for input channels (PI), for output channels (PO) and input/output channels (PIO) to construct index sets. Indices of the sets are selected randomly using a hypergeometric distribution without replacement at the initialization of a training step, and fixed in the rest of the training. Implementation details and experimental analyses are given in the supp. mat.

## 4  Bounding Generalization Errors using Fine-grained Weights

Mathematically, norms of concatenated weights $\omega_{g,l}$, $\forall g$, are lower bounded by products of norms of component weights $\omega_{g,l}^i$, $\forall i$. Weights are rescaled dynamically at each $t^{th}$ epoch of an optimization method proposed to train DNNs using $\mathfrak{R}_{i,l}^t = \frac{\gamma_{i,l}}{\lambda_{i,l}^t}$, where $\gamma_{i,l} > 0$ is a geometric scaling parameter and $\lambda_{i,l}^t$ is the standard deviation of features input to the $i^{th}$ weight in the $g^{th}$ group $\omega_{g,l}^i$, $\forall i, g$.

The scaling parameter $\mathfrak{R}_{i,l}^t$ enables us to upper bound the norms of weights by 1 (see Table 1). Suppose that all layers have the same width $\varpi$, weights have the same length $\mathcal{K}$ and the same stride $\mathfrak{s}$. Then, generalization bounds are obtained for DNNs using these fixed parameters by $\|\omega_l\|_2 = \frac{\mathcal{K}}{\mathfrak{s}}$, $\|\omega_l\|_F = \sqrt{\varpi}$ and $\|\omega_l\|_{2\to1} = \varpi$. We compute a concatenated weight matrix $\omega_{g,l} = (\omega_{g,l}^1, \omega_{g,l}^2, \ldots, \omega_{g,l}^{|\mathfrak{g}|})$ for the $g^{th}$ weight group of size $|\mathfrak{g}|$, $g = 1, 2, \ldots, G_l$, $\forall l$ using a weight grouping strategy. Then, we have upper bounds of norms by $\|\omega_{g,l}\|_F \le \delta_{g,l,F} \le 1$, $\|\omega_{g,l}\|_2 \le \delta_{g,l,2} \le 1$ and $\|\omega_{g,l}\|_{2\to1} \le \delta_{g,l,2\to1} \le 1$, $g = 1, 2, \ldots, G_l$, which are defined in Table 2.

## 4.1 The Proof strategy for Computation of the Upper Bounds

The upper bounds are computed in Proposition 1 in the supplemental material. The proof strategy is summarized as follows:

- Let $\mathfrak{b}_{i,l}$ be multiplication of the number of input channels and the size of the receptive field of the unit that employs $\omega_{g,l}^i$, and $\hat{\mathfrak{b}}_{i,l}$ be multiplication of the dimension of output feature maps and the number of output channels used at the $l^{th}$ layer, respectively. Then, geometric scaling $\gamma_{i,l}$ of the weight space of $\omega_{g,l}^i$ is computed by

$$\gamma_{i,l} = \sqrt{\frac{1}{\mathfrak{b}_{i,l} + \hat{\mathfrak{b}}_{i,l}}}. \tag{2}$$

- We can consider that standard deviation of features satisfy $\lambda_{i,l}^t \geq 1$ using two approaches. First, by employing the central limit theory for weighted summation of random variables of features, we can prove that $\lambda_{i,l}^t$ converges to 1 asymptotically, as popularly employed in the previous works. Second, we can assume that we apply batch normalization (BN) by setting the re-scaling parameter of the BN to 1. Thereby, we can obtain $\frac{1}{\lambda_{i,l}^t} \leq 1$. By definition, $\gamma_{i,l}^2 < B_l, \forall i, l$. In order to show that $\sigma(\omega_{g,l}^i) \leq (\gamma_{i,l})^{-1}, \forall i, l$, we apply the Bai-Yin law [25, 26]. Thereby, we conclude that norms of concatenated weights belonging to groups given in Table 1 are upper bounded by 1, if the corresponding component weights given in Table 2 are rescaled by $\mathfrak{R}_{i,l}^t, \forall i, l, t$ during training.

*Remark* 1. We compute norms of weights belonging to each different manifold in Table 2. Following Proposition 1 given in the supplemental material, we have $\|\omega_{g,l}\|_F \geq (\prod_{i=1}^{|\mathfrak{g}|} \|\omega_{g,l}^i\|_F)^{1/|\mathfrak{g}|}$, $\|\omega_{g,l}\|_2 \geq (\prod_{i=1}^{|\mathfrak{g}|} \|\omega_{g,l}^i\|_2)^{1/|\mathfrak{g}|}$ and $\|\omega_{g,l}\|_{2\to1} \geq (\prod_{i=1}^{|\mathfrak{g}|} \|\omega_{g,l}^i\|_{2\to1})^{1/|\mathfrak{g}|}$.

## 4.2 Generalization of Scaled Weight Initialization Methods

Note that scaling by $\mathfrak{R}_{i,l}^t$ computed using (2) is different from the scaling method suggested in [12] such that our proposed method assures tighter upper bound for norms of weights. Our method also generalizes the scaling method given in [27] (a.k.a. *Xavier initialization*)in two ways. First, we use size of input receptive fields and output feature spaces which determine dimension of weight manifolds, as well as number of input and output dimensions which determine number of manifolds used in groups.

Second, we perform scaling not just at initialization but also at each $t^{th}$ epoch of the optimization method. Therefore, diversity of weights is controlled and we can obtain weights uniformly distributed on the corresponding manifolds whose geometric properties change dynamically at each epoch. Applying this property with the results given in [20], we can prove that NNs applying the proposed scaling have no spurious local minima[4]. In addition, our method generalizes the scaling method proposed in [22] for weight normalization by incorporating geometric properties of weight manifolds.

## 5 Optimization using Fine-Grained Stochastic Gradient Descent in DNNs

In this section, we address the problem of optimization of DNNs considering constraints on weight matrices according the upper bounds on their norms given in Section 4.

### 5.1 Optimization on POMs in DNNs: Challenges

Employment of a vanilla SGD on POMs with assurance to convergence to local or global minima for training DNNs using back-propagation (BP) with collections of POMs is challenging. More precisely, we observe early divergence of SGD, and exploding and vanishing gradients in practice, due to the following theoretical properties of collections of POMs:

**Algorithm 1** Optimization using FG-SGD on products manifolds of fine-grained weights.

1: **Input:** $T$ (number of iterations), $S$ (training set),
   $\Theta$ (set of hyperparameters), $\mathcal{L}$ (a loss function), $\mathcal{I}_g^l \subseteq \mathcal{I}_{\mathcal{G}_l}, \forall g, l$.
2: **Initialization:** Construct a collection of products of weight manifolds $\mathcal{G}_l$, initialize re-scaling
   parameters $\mathcal{R}_l^t$ and initialize weights $\omega_{g,l}^t \in \mathbb{M}_{g,l}$ with $\mathcal{I}_g^l \subseteq \mathcal{I}_{\mathcal{G}_l}, \forall m, l$.
3: **for** each iteration $t = 1, 2, \ldots, T$ **do**
4:     **for** each layer $l = 1, 2, \ldots, L$ **do**
5:         $\mathrm{grad}\mathcal{L}(\omega_{g,l}^t) := \Pi_{\omega_{g,l}^t}\Big(\mathrm{grad}_E \mathcal{L}(\omega_{g,l}^t), \Theta, \mathcal{R}_l^t\Big), \forall \mathcal{G}_l.$
6:         $v_t := h(\mathrm{grad}\mathcal{L}(\omega_{g,l}^t), r(t, \Theta)), \forall \mathcal{G}_l.$
7:         $\omega_{g,l}^{t+1} := \phi_{\omega_{g,l}^t}(v_t, \mathcal{R}_l^t), \forall \omega_{g,l}^t, \forall \mathcal{G}_l.$
8:     **end for**
9: **end for**
10: **Output:** A set of estimated weights $\{\omega_{g,l}^T\}_{l=1}^L, \forall g$.

---

• Geometric properties of a POM $\mathbb{M}_{g,l}$ can be different from those of its component manifolds $\mathbb{M}_\iota$, even if the component manifolds are identical. For example, we observe locally varying curvatures when we construct POMs of unit spheres. Weight manifolds with more complicated geometric properties can be obtained using the proposed PIO strategy, especially by constructing collections of POMs of non-identical manifolds. Therefore, assumption on existence of compact weight subsets in POMs may fail due to locally varying metrics within a nonlinear component manifold and among different component manifolds.

• When we optimize weights using SGD in DNNs, we first obtain gradients computed for each weight $\omega_{g,l} \in \mathbb{M}_{g,l}$ at the $l^{th}$ layer from the $(l+1)^{st}$ layer using backpropagation (BP). Then, each weight $\omega_{g,l}$ moves on $\mathbb{M}_{g,l}$ according to the gradient. However, curvatures and metrics of $\mathbb{M}_{g,l}$ can locally vary, and they may be different from those of component manifolds of $\mathbb{M}_{g,l}$ as explained above.

This geometric drawback causes two critical problems. First, weights can be moved incorrectly if we move them using only gradients computed for each individual component of the weights, as popularly employed for the Euclidean linear weight spaces. Second, due to incorrect employment of gradients and movement of weights, probability of failure of the SGD cannot be bounded, and convergence cannot be achieved (see proofs of Theorem 2, Corollary 1 and Corollary 2 for details). In practice, this causes unbounded increase or decrease of values of gradients and weights.

## 5.2 A Geometric Approach to Optimization on POMs in DNNs

In order to address these problems for training DNNs, we first analyze the relationship between geometric properties of POMs and those of their component manifolds in the next theorem.
*Remark* 2. (See Lemma 1 given in the supp. mat. for the complete proof of the following propositions) Our main theoretical results regarding geometric properties of POMs are summarized as follows:

1. **Computation of metrics:** A metric defined on a product weight manifold $\mathbb{M}_{g,l}$ can be computed by superposition (i.e. linear combination) of Riemannian metrics of its component manifolds.

2. **Lower bounds of sectional curvatures:** Sectional curvature of a product weight manifold $\mathbb{M}_{g,l}$ is lower bounded by 0. ∎

### 5.2.1 Development of FG-SGD employing Geometry of POMs

We use the first result (1) for *projection* of Euclidean gradients obtained using BP onto product weight manifolds. More precisely, we can compute *norms of gradients* at weights on a product weight manifold by linear superposition of those computed on its component manifolds in FG-SGD. Thereby, we can move a weight on a product weight manifold by (i) retraction of components of the weight on component manifolds of the product weight manifold, and (ii) concatenation of projected weight components in FG-SGD.

The second result (2) show that some sectional curvatures vanish on a product weight manifold $\mathbb{M}_{g,l}$. For instance, suppose that each component weight manifold $\mathcal{M}_{\iota,l}$ of $\mathbb{M}_{g,l}$ is a unit two-sphere $\mathbb{S}^2$,

$\forall \iota \in \mathcal{I}_{\mathcal{G}_l}$. Then, $\mathbb{M}_{g,l}$ has unit curvature along two-dimensional subspaces of its tangent spaces, called two-planes. However, $\mathbb{M}_{g,l}$ has zero curvature along all two-planes spanning exactly two distinct spheres. In addition, weights can always move according to a non-negative bound on sectional curvature of compact product weight manifolds on its tangent spaces. Therefore, we do not need to worry about varying positive and negative curvatures observed at its different component manifolds. The second result also suggests that learning rates need to be computed adaptively by a function of *norms of gradients* and *bounds on sectional curvatures* at each layer of the DNN and at each epoch of FG-SGD for each weight $\omega$ on each product weight manifold $\mathbb{M}_{g,l}$. We employ these results to analyze convergence of FG-SGD and compute its adaptive step size in the following sections.

### 5.2.2 Optimization on POMs using FG-SGD in DNNs

An algorithmic description of our proposed fine-grained SGD (FG-SGD) is given in Algorithm 1. At the initialization of the FG-SGD, we identify the component weight manifolds $\mathcal{M}_{\iota,l}$ of each product weight manifold $\mathbb{M}_{g,l}$ according to the constraints that will be applied on the weights $\omega_\iota \in \mathcal{M}_{\iota,l}$ for each $g^{th}$ group at each $l^{th}$ layer. For $t = 1$, each manifold $\mathcal{M}_{\iota,l}$ is scaled by $\mathfrak{R}_{\iota,l}^{t=1}$ using $\lambda_{\iota,l}^{t=1} = 1, \forall \iota, l$. For $t > 1$, each $\mathcal{M}_{\iota,l}$ is re-scaled by $\mathfrak{R}_{\iota,l}^t \in \mathcal{R}_l^t$ computing empirical standard deviation $\lambda_\iota^t$ of features input to each weight of $\mathcal{M}_{\iota,l}$, and $\mathcal{R}_l^t$ is the set of all re-scaling parameters computed at the $t^{th}$ epoch at each $l^{th}$ layer. When we employ a FG-SGD on a product weight manifold $\mathbb{M}_{g,l}$ each weight $\omega_{g,l}^t \in \mathbb{M}_{g,l}$ is moved on $\mathbb{M}_{g,l}$ in the descent direction of gradient of loss at each $t^{th}$ step of the FG-SGD by the following steps:

**Line 5 (Projection of gradients on tangent spaces):** The gradient $\mathrm{grad}_E \, \mathcal{L}(\omega_{g,l}^t)$, obtained using back-propagation from the upper layer, is projected onto the tangent space $\mathcal{T}_{\omega_{g,l}^t} \mathbb{M}_{g,l} = \underset{\iota \in \mathcal{I}_g^l}{\times} \mathcal{T}_{\omega_{\iota,l}^t} \mathbb{M}_{\iota,l}$ to compute $\mathrm{grad}\mathcal{L}(\omega_{g,l}^t)$ at the weight $\omega_{g,l}^t$ using the results given in Remark 2, where $\mathcal{T}_{\omega_{\iota,l}^t} \mathbb{M}_{\iota,l}$ is the tangent space at $\omega_{\iota,l}^t$ on the component manifold $\mathbb{M}_{\iota,l}$ of $\mathbb{M}_{g,l}$.

**Line 6 (Movement of weights on tangent spaces):** The weight $\omega_{g,l}^t$ is moved on $\mathcal{T}_{\omega_{g,l}^t} \mathbb{M}_{g,l}$ using

$$h(\mathrm{grad}\mathcal{L}(\omega_{g,l}^t), r(t,\Theta)) = -\frac{r(t,\Theta)}{\iota(\omega_{g,l}^t)} \mathrm{grad}\mathcal{L}(\omega_{g,l}^t), \tag{3}$$

where $r(t,\Theta)$ is the learning rate that satisfies

$$\sum_{t=0}^{\infty} r(t,\Theta) = +\infty \text{ and } \sum_{t=0}^{\infty} r(t,\Theta)^2 < \infty, \tag{4}$$

$$\iota(\omega_{G_l^m}^t) = \max\{1, \Gamma_1^t\}^{\frac{1}{2}} \tag{5}$$

$\Gamma_1^t = (R_{g,l}^t)^2 \Gamma_2^t$, $\qquad R_{g,l}^t \triangleq \|\mathrm{grad}\mathcal{L}(\omega_{g,l}^t)\|_2$ is computed using (6), $\Gamma_2^t = \max\{(2\rho_{g,l}^t + R_{g,l}^t)^2, (1 + \mathfrak{c}_{g,l}(\rho_{g,l}^t + R_{g,l}^t))\}$, $\mathfrak{c}_{g,l}$ is the sectional curvature of $\mathbb{M}_{g,l}$, $\rho_{g,l}^t \triangleq \rho(\omega_{g,l}^t, \hat{\omega}_{g,l})$ is the geodesic distance between $\omega_{g,l}^t$ and a local minima $\hat{\omega}_{g,l}$ on $\mathbb{M}_{g,l}$.

The following result is used for computation of the $\ell_2$ norm of gradients.

***Theorem* 1** (Computation of gradients on tangent spaces). The $\ell_2$ norm $\|\mathrm{grad}\mathcal{L}(\omega_{g,l}^t)\|_2$ of the gradient $\mathrm{grad}\mathcal{L}(\omega_{g,l}^t)$ residing on $\mathcal{T}_{\omega_{g,l}^t} \mathbb{M}_{g,l}$ at the $t^{th}$ epoch and the $l^{th}$ layer can be computed by

$$\|\mathrm{grad}\mathcal{L}(\omega_{g,l}^t)\|_2 = \Big( \sum_{\iota \in \mathcal{I}_g^l} \mathrm{grad}\mathcal{L}(\omega_{\iota,l}^t)^2 \Big)^{\frac{1}{2}}, \tag{6}$$

where $\mathrm{grad}\mathcal{L}(\omega_{\iota,l}^t)$ is the gradient computed for $\omega_{\iota,l}^t$ on the tangent space $\mathcal{T}_{\omega_{\iota,l}^t} \mathbb{M}_\iota, \forall \iota \in \mathcal{I}_g^l$. ∎

We compute norms of gradients on tangent spaces of product manifolds by just superposing gradients computed on tangent spaces of component manifolds using (6). Norms of gradients, section curvatures of the product manifolds and their metric properties affect convergence of weights on the product manifolds to local or global minima. More precisely, they define an upper bound on the divergence of the weights from local or global minima (please see proof of Theorem 2 in the supplemental material for details). This upper bound is further bounded by normalizing gradients in (3).

**Line 7 (Projection of moved weights onto product of manifolds):** The moved weight located at $v_t$ is projected onto $\mathbb{M}_{g,l}$ re-scaled by $\mathcal{R}_l^t$ using $\phi_{\omega_{g,l}^t}(v_t, \mathcal{R}_l^t)$ to compute $\omega_{g,l}^{t+1}$, where $\phi_{\omega_{g,l}^t}(v_t, \mathcal{R}_l^t)$ is an exponential map, or a retraction, i.e. an approximation of the exponential map [28]. The function $\imath(\omega_{g,l}^t)$ used for computing step size in (3) is employed as a regularizer to control the change of gradient $\mathrm{grad}\mathcal{L}(\omega_{g,l}^t)$ at each step of FG-SGD. This property is examined in the experimental analyses in the supp. mat. For computation of $\imath(\omega_{g,l}^t)$, we use (6) with Theorem **1**. In FG-SGD, weights residing on each POM are moved and projected jointly on the POMs, by which we can employ their interaction using the corresponding gradients considering nonlinear geometry of manifolds unlike SGD methods studied in the literature. G-SGD can consider interactions between component manifolds as well as those between POMs in groups of weights. Employment of (3) and (4) at line 7, and retractions at line 8 are essential for assurance of convergence as explained next.

### 5.3 Convergence Properties of FG-SGD

Convergence properties of the proposed FG-SGD used to train DNNs are summarized as follows:

**Convergenge to local minima:** The loss function of a non-linear DNN, which employs the proposed FG-SGD, converges to a local minimum, and the corresponding gradient converges to zero almost surely (a.s.). The formal theorem and proof are given in Theorem 2 in the supplemental material.

**Convergenge to global minima:** Loss functions of particular DNNs such as linear DNNs, one-hidden-layer CNNs, one-hidden-layer Leaky Relu networks, nonlinear DNNs with specific network structures (e.g. pyramidal networks), trained using FG-SGD, converge to a global minimum a.s. under mild assumptions on data (e.g. being distributed from Gaussian distribution, normalized, and realized by DNNs). The formal theorem and proof of this result are given in Corollary 1 in the supp. mat. The proof idea is to use the property that local minima of loss functions of these networks are global minima under these assumptions, by employing the results given in the recent works [29–38].

**An example for adaptive computation of step size:** Suppose that $\mathbb{M}_\iota$ are identified by $n_\iota \geq 2$ dimensional unit sphere, or the sphere scaled by the proposed scaling method. If step size is computed using (3) with

$$\imath(\omega_{G_l^m}^t) = \left(\max\{1, (R_{G_l^m}^t)^2(2 + R_{G_l^m}^t)^2\}\right)^{\frac{1}{2}}, \tag{7}$$

then the loss function converges to local minima for a generic class of nonlinear DNNs, and to global minima for DNNs characterized in Corollary 1. The formal theorem and proof of this result are given in Corollary 2 in the supp. mat. The proof idea follows the property that the unit sphere has positive sectional curvature, and a product of unit spheres has non-negative curvature.

Using these results for the sphere, the normalizing function (5) is computed by the maximum of 1 and a polynomial function of the norm of gradients (6). Note that, variations of this function (7) have been used in practice to train large DNNs *successfully*, i.e. avoiding exploding and vanishing gradients. Our mathematical framework elucidates the underlying theory behind the success of gradient normalization methods for convergence of DNNs to local and global minima.

## 6 Conclusion and Discussion

We introduced and elucidated a problem of training CNNs using multiple constraints employed on convolution weights with convergence properties. Following our theoretical results, we proposed the FG-SGD algorithm and adaptive step size estimation methods for optimization on collections of POMs that are identified by the constraints. Due to page limit, experimental analyses are given in the supplemental material. In these analyses, we observe that our proposed methods can improve convergence properties and classification performance of CNNs. Overall, the results show that employment of collections of POMs using FG-SGD can boost the performance of various different CNNs on various datasets. We consider a research direction for investigating how far local minima are from global minima in search spaces of FG-SGD using products of weight manifolds with nonlinear DNNs and their convergence rates.

We believe that our proposed mathematical framework and results will be useful and inspiring for researchers to study geometric properties of parameter spaces of deep networks, and to improve our understanding of deep feature representations.

## Footnotes

[2]We use shorthand notation for matrix concatenation such that $[W_{c,d,l}]_{c=1}^{C_l} \triangleq [W_{1,d,l}, W_{2,d,l}, \cdots, W_{C_l,d,l}]$.

[3]In this work, we consider Riemannian manifolds of normalized weights defined in the previous section. Formal definitions are given in the supp. mat.

[4]We omit the formal theorem and the proof on this result in this work to focus on our main goal and novelty for optimization with multiple weight manifolds.

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
