[Supplementary Material]

# Supplemental Material for Fine-grained Optimization of Deep Neural Networks

## Abstract

In this document, supplemental material of the paper titled "Fine-grained Optimization of Deep Neural Networks" is provided. We first introduce proofs of the theorems in the following section. Next, we first provide the manifolds and the maps used in implementation of the algorithm. Then, we provide implementation details of algorithms used in experimental analyses whose results are given in the main text. Additional results are given in the last section.

## 1 Bounding Generalization Errors using Fine-grained Weights

***Proposition* 1** (Bounding norms of weight matrices and generalization errors of DNNs). Suppose that DNNs given in Table 3 are trained using weights renormalized by the renormalization method proposed in the main text according to the Frobenius, spectral and column/row wise norms with reparameterization parameters $\mathfrak{R}_{i,l}^{t}, \forall i, l, t$ with $\lambda_{i,l}^{t} \geq 1$. Then, norms of renormalized weight matrices are upper bounded by a constant number, and generalization errors of the corresponding DNNs are asymptotically bounded as given in the rightmost column of the Table 3, denoted by **DNNs** (our proposed reparameterization).

*Proof.* Suppose that matrices of weights $\omega_{g,l}^{i} \in \mathbb{R}^{A_l \times B_l}$ belonging to the $g^{th}$ group of size $|\mathfrak{g}|, g = 1, 2, \ldots, G_l, \forall l$ have the same size $A_l \times B_l$ for simplicity, and $\sigma(\omega_{g,l}^{i})$ denotes the top singular value of $\omega_{g,l}^{i}$. Let $\|\omega_{g,l}^{i}\|_F$, $\|\omega_{g,l}^{i}\|_2$, and $\|\omega_{g,l}^{i}\|_{2 \to 1}$, denote respectively the Frobenius, spectral and $\ell_{2 \to 1}$ norms of the weight $\omega_{g,l}^{i}$.

We note that, matrices of weights $\omega_{g,l}^{i}$ belonging to the $g^{th}$ group are concatenated by $\omega_{g,l} = (\omega_{g,l}^{1}, \omega_{g,l}^{2}, \ldots, \omega_{g,l}^{\mathfrak{g}}), \forall g = 1, 2, \ldots, G_l$, to perform group-wise operations in DNNs. Thereby, we can employ bounds for norms of each concatenated matrix in generalization error bounds given in the leftmost column of Table 3, denoted by **DNNs** (bounds on norms), and obtain the bounds given in the rightmost column of the Table 3, denoted by **DNNs**(our proposed reparameterization).

We compute norms of matrices of normalized weights $\omega_{g,l}^{i}$ belonging to each different manifold in Table 1. These norms are computed using simple matrix calculus considering definitions of matrices residing on each manifold according to the definition given in Table 2. From these calculations given in Table 1, we observe that, the maximum of norm values that a weight $\omega_{g,l}^{i}$ belonging to the sphere can achieve is $\mathbb{M}_{sp}(\omega_{g,l}^{i}) = \sigma(\omega_{g,l}^{i})$, that of a weight belonging to the Stiefel manifold is $\mathbb{M}_{st}(\omega_{g,l}^{i}) = (B_l)^{1/2}$, and that of a weight belonging to the oblique manifold is $\mathbb{M}_{ob}(\omega_{g,l}^{i}) = \max\{(B_l)^{1/2}, \sigma(\omega_{g,l}^{i})\}$.

In our proposed renormalization method, we first normalize each weight matrix such that the norm of the matrix $\omega_{g,l}^{i}$ can have one of these values $\mathbb{M}_{sp}(\omega_{g,l}^{i})$, $\mathbb{M}_{st}(\omega_{g,l}^{i})$ and $\mathbb{M}_{ob}(\omega_{g,l}^{i})$. Therefore, we need to reparameterize weight matrices such that norm of each reparameterized weight is less than 1.0. For this purpose, we need show that the rescaling of these norm values by $\mathfrak{R}_{i,l}^{t}$ is upper bounded by 1.0.

Weights are rescaled dynamically at each $t^{th}$ epoch of an optimization method proposed to train DNNs using $\mathfrak{R}_{i,l}^t = \frac{\gamma_{i,l}^t}{\lambda_{i,l}^t}$, where $0 < \gamma_{i,l} < 1.0$ is a geometric scaling parameter and $\lambda_{i,l}^t$ is the standard deviation of features input to the $i^{th}$ weight in the $g^{th}$ group $\omega_{g,l}^i$, $\forall i, g$. By assumption, $\lambda_{i,l}^t \leq 1.0, \forall i, t, l$. By definition, $B_l \gamma_{i,l}^2 \leq 1.0, \forall i, l$. In order to show that $\sigma(\omega_{g,l}^i) \leq (\gamma_{i,l})^{-1}, \forall i, l$, we apply the Bai-Yin law [1, 2]. Thereby, we conclude that norms of concatenated weights belonging to groups given in Table 3 are upper bounded by 1, if the corresponding component weights given in Table 1 are rescaled by $\mathfrak{R}_{i,l}^t$, $\forall i, l, t$ during training of DNNs.

Since norm of each weight matrix $\omega_{g,l}^i$ is bounded by 1.0, their multiplication for all $g = 1, 2, \ldots, G_l$ and $\forall l$ is also bounded by 1.0.

$\square$

Table 1: Comparison of norms of weights belonging to different weight manifolds.

| Norms | (i) Sphere | (ii) Stiefel | (iii) Oblique |
|---|---|---|---|
| $\|\omega_{g,l}^i\|_2$ | $\sigma(\omega_{g,l}^i)$ | 1.0 | $\sigma(\omega_{g,l}^i)$ |
| $\|\omega_{g,l}^i\|_F$ | 1.0 | $(B_l)^{1/2}$ | $(B_l)^{1/2}$ |
| $\|\omega_{g,l}^i\|_{2 \to 1}$ | 1.0 | $(B_l)^{1/4}$ | $(B_l)^{1/4}$ |

Table 2: Embedded weight manifolds $\mathcal{M}_\iota$ used for construction of collection of POMs $\mathbb{M}_{G_l}$, $\forall l$, in the experimental analyses. The Frobenius norm of a convolution weight $\omega$ is denoted by $\|\omega\|_F$. The $b^{th}$ column vector of a weight matrix $\omega \in \mathbb{R}^{A_l \times B_l}$ is denoted by $\omega_b$. An $B_l \times B_l$ identity matrix is denoted by $I_{B_l}$.

| Manifolds | Definitions |
|---|---|
| The Sphere | $\mathcal{S}(A_l, B_l) = \{\omega \in \mathbb{R}^{A_l \times B_l} : \|\omega\|_F = 1\}$ |
| The Oblique | $\mathcal{OB}(A_l, B_l) = \{\omega \in \mathbb{R}^{A_l \times B_l} : \|\omega_b\|_F = 1, \forall b = 1, 2, \ldots, B_l\}$ |
| The Stiefel | $St(A_l, B_l) = \{\omega \in \mathbb{R}^{A_l \times B_l} : (\omega^{\mathrm{T}} \omega) = I_{B_l}\}$ |

## 2 Proofs of Theorems given in the Main Text

**Definition 2.1** (Sectional curvature of component manifolds)**.** Let $\mathfrak{X}(\mathcal{M}_\iota)$ denote the set of smooth vector fields on $\mathcal{M}_\iota$. The sectional curvature of $\mathcal{M}_\iota$ associated with a two dimensional subspace $\mathfrak{T} \subset \mathcal{T}_{\omega_\iota} \mathcal{M}_\iota$ is defined by

$$\mathfrak{c}_\iota = \frac{\langle \mathcal{C}_\iota(X_{\omega_\iota}, Y_{\omega_\iota}) Y_{\omega_\iota}, X_{\omega_\iota} \rangle}{\langle X_{\omega_\iota}, X_{\omega_\iota} \rangle \langle Y_{\omega_\iota}, Y_{\omega_\iota} \rangle - \langle X_{\omega_\iota}, Y_{\omega_\iota} \rangle^2} \tag{1}$$

where $\mathcal{C}_\iota(X_{\omega_\iota}, Y_{\omega_\iota}) Y_{\omega_\iota}$ is the Riemannian curvature tensor, $\langle \cdot, \cdot \rangle$ is an inner product, $X_{\omega_\iota} \in \mathfrak{X}(\mathcal{M}_\iota)$ and $Y_{\omega_\iota} \in \mathfrak{X}(\mathcal{M}_\iota)$ form a basis of $\mathfrak{T}$. $\blacksquare$

**Definition 2.2** (Riemannian connection on component embedded weight manifolds)**.** Let $\mathfrak{X}(\mathcal{M}_\iota)$ denote the set of smooth vector fields on $\mathcal{M}_\iota$ and $\mathfrak{F}(\mathcal{M}_\iota)$ denote the set of smooth scalar fields on $\mathcal{M}_\iota$. The Riemannian connection $\bar{\nabla}$ on $\mathcal{M}_\iota$ is a mapping [6]

$$\bar{\nabla} : \mathfrak{X}(\mathcal{M}_\iota) \times \mathfrak{X}(\mathcal{M}_\iota) \to \mathfrak{X}(\mathcal{M}_\iota) : (X_{\omega_\iota}, Y_{\omega_\iota}) \mapsto \bar{\nabla} X_{\omega_\iota} Y_{\omega_\iota} \tag{2}$$

which satisfies the following properties:

Table 3: Comparison of generalization bounds. $\mathcal{O}$ denotes big-O and $\tilde{\mathcal{O}}$ is soft-O. $\delta_{l,F}$, $\delta_{l,2}$, and $\delta_{l,2\to 1}$ denotes upper bounds of the Frobenius norm $\|\omega_l\|_F \le \delta_{l,F}$, spectral norm $\|\omega_l\|_2 \le \delta_{l,2}$ and the sum of the Euclidean norms for all rows $\|\omega_l\|_{2\to 1} \le \delta_{l,2\to 1}$ ($\ell_{2\to 1}$) of weights $\omega_l$ at the $l^{th}$ layer of an $L$ layer DNN using $N$ samples. Suppose that all layers have the same width $\varpi$, weights have the same length $\mathcal{K}$ and the same stride $\mathfrak{s}$. Then, generalization bounds are obtained for DNNs using these fixed parameters by $\|\omega_l\|_2 = \frac{\mathcal{K}}{\mathfrak{s}}$, $\|\omega_l\|_F = \sqrt{\varpi}$ and $\|\omega_l\|_{2\to 1} = \varpi$. We compute a concatenated weight matrix $\omega_{g,l} = (\omega_{g,l}^1, \omega_{g,l}^2, \ldots, \omega_{g,l}^{|\mathfrak{g}|})$ for the $g^{th}$ weight group of size $|\mathfrak{g}|$, $g = 1, 2, \ldots, G_l, \forall l$ using a weight grouping strategy. Then, we have upper bounds of norms by $\|\omega_{g,l}\|_F \le \delta_{g,l,F} \le 1$, $\|\omega_{g,l}\|_2 \le \delta_{g,l,2} \le 1$ and $\|\omega_{g,l}\|_{2\to 1} \le \delta_{g,l,2\to 1} \le 1$, $g = 1, 2, \ldots, G_l$, which are defined in Table 1.

|  | **DNNs (dynamic group scaling)** |
|---|---|
| Neyshabur et al. [3] | $\mathcal{O}\left( \dfrac{2^L \prod\limits_{l=1}^{L} \prod\limits_{g=1}^{G_l} \delta_{g,l,F}}{\sqrt{N}} \right)$ |
| Bartlett et al. [4] | $\tilde{\mathcal{O}}\left( \dfrac{\prod\limits_{l=1}^{L} \prod\limits_{g=1}^{G_l} \delta_{g,l,2}}{\sqrt{N}} \left( \sum\limits_{l=1}^{L} \prod\limits_{g=1}^{G_l} \left(\dfrac{\delta_{g,l,2\to 1}}{\delta_{g,l,2}}\right)^{\frac{2}{3}} \right)^{\frac{3}{2}} \right)$ |
| Neyshabur et al. [5] | $\tilde{\mathcal{O}}\left( \dfrac{\prod\limits_{l=1}^{L} \prod\limits_{g=1}^{G_l} \delta_{g,l,2}}{\sqrt{N}} \sqrt{L^2 \varpi \sum\limits_{l=1}^{L} \prod\limits_{g=1}^{G_l} \dfrac{\delta_{g,l,F}^2}{\delta_{g,l,2}^2}} \right)$ |

1. $\bar{\nabla}_{pX_{\omega_\iota}+qY_{\omega_\iota}} Z_{\omega_\iota} = p\bar{\nabla} Z_{\omega_\iota} + q\nabla_{Y_{\omega_\iota}} Z_{\omega_\iota}$,

2. $\bar{\nabla} X_{\omega_\iota}(\alpha Y_{\omega_\iota} + \beta Z_{\omega_\iota}) = \alpha\bar{\nabla}_{X_{\omega_\iota} Y_{\omega_\iota}} + \beta\bar{\nabla}_{X_{\omega_\iota}} Z_{\omega_\iota}$,

3. $\bar{\nabla}_{X_{\omega_\iota}}(pY_{\omega_\iota}) = (X_{\omega_\iota}p)Y_{\omega_\iota} + p\bar{\nabla}_{X_{\omega_\iota}} Y_{\omega_\iota}$,

4. $\bar{\nabla}_{X_{\omega_\iota}} Y_{\omega_\iota} - \bar{\nabla}_{Y_{\omega_\iota}} X_{\omega_\iota} = [X_{\omega_\iota}, Y_{\omega_\iota}]$ and

5. $Z_{\omega_\iota}\langle X_{\omega_\iota}, Y_{\omega_\iota}\rangle = \langle \bar{\nabla}_{Z_{\omega_\iota}} X_{\omega_\iota}, Y_{\omega_\iota}\rangle + \langle X_{\omega_\iota}, \bar{\nabla}_Z Y_{\omega_\iota}\rangle$

where $X_{\omega_\iota}, Y_{\omega_\iota}, Z_{\omega_\iota} \in \mathfrak{X}(\mathcal{M}_\iota)$, $p, q \in \mathfrak{F}(\mathcal{M}_\iota)$, $\alpha, \beta \in \mathbb{R}$, $\langle \cdot, \cdot \rangle$ is an inner product, $[X_{\omega_\iota}, Y_{\omega_\iota}]$ is the Lie bracket of $X_{\omega_\iota}$ and $Y_{\omega_\iota}$, and defined by $[X_{\omega_\iota}, Y_{\omega_\iota}]p = X_{\omega_\iota}(Y_{\omega_\iota}p) - Y_{\omega_\iota}(X_{\omega_\iota}p)$, $\forall p \in \mathfrak{F}(\mathcal{M}_\iota)$.

***Lemma*** 1 (Metric and curvature properties of POMs). Suppose that $u_\iota \in \mathcal{T}_{\omega_\iota}\mathcal{M}_\iota$ and $v_\iota \in \mathcal{T}_{\omega_\iota}\mathcal{M}_\iota$ are tangent vectors belonging to the tangent space $\mathcal{T}_{\omega_\iota}\mathcal{M}_\iota$ computed at $\omega_\iota \in \mathcal{M}_\iota$, $\forall \iota \in \mathcal{I}_{G_l}$. Then, tangent vectors $u_{G_l} \in \mathcal{T}_{\omega_{G_l}}\mathbb{M}_{G_l}$ and $v_{G_l} \in \mathcal{T}_{\omega_{G_l}}\mathbb{M}_{G_l}$ are computed at $\omega_{G_l} \in \mathbb{M}_{G_l}$ by concatenation as $u_{G_l} = (u_1, u_2, \cdots, u_{|\mathcal{I}_{G_l}|})$ and $v_{G_l} = (v_1, v_2, \cdots, v_{|\mathcal{I}_{G_l}|})$. If each weight manifold $\mathcal{M}_\iota$ is endowed with a Riemannian metric $\mathfrak{d}_\iota$, then a $G_l$-POM is endowed with the metric $\mathfrak{d}_{G_l}$ computed by

$$\mathfrak{d}_{G_l}(u_{G_l}, v_{G_l}) = \sum_{\iota \in \mathcal{I}_{G_l}} \mathfrak{d}_\iota(u_\iota, v_\iota). \tag{3}$$

In addition, suppose that $\bar{C}_\iota$ is the Riemannian curvature tensor field (endomorphism) [7] of $\mathcal{M}_\iota$, $x_\iota, y_\iota \in \mathcal{T}_{\omega_\iota}\mathcal{M}_\iota$, $\forall \iota \in \mathcal{I}_{G_l}$ defined by

$$\bar{C}_\iota(u_\iota, v_\iota, x_\iota, y_\iota) = \langle C_\iota(U, V)X, Y\rangle_{\omega_\iota}, \tag{4}$$

where $U, V, X, Y$ are vector fields such that $U_{\omega_\iota} = u_\iota$, $V_{\omega_\iota} = v_\iota$, $X_{\omega_\iota} = x_\iota$, and $Y_{\omega_\iota} = y_\iota$. Then, the Riemannian curvature tensor field $\bar{C}_{G_l}$ of $\mathbb{M}_{G_l}$ is computed by

$$\bar{C}_{G_l}(u_{G_l}, v_{G_l}, x_{G_l}, y_{G_l}) = \sum_{\iota \in \mathcal{I}_{G_l}} \bar{C}_\iota(u_\iota, v_\iota, x_\iota, y_\iota), \tag{5}$$

where $x_{G_l} = (x_1, x_2, \cdots, x_{|\mathcal{I}_{G_l}|})$ and $y_{G_l} = (y_1, y_2, \cdots, y_{|\mathcal{I}_{G_l}|})$. Moreover, $\mathbb{M}_{G_l}$ has never strictly positive sectional curvature $\mathfrak{c}_{G_l}$ in the metric (3). In addition, if $\mathbb{M}_{G_l}$ is compact, then $\mathbb{M}_{G_l}$ does not admit a metric with negative sectional curvature $\mathfrak{c}_{G_l}$. ∎

*Proof.* Since each weight manifold $\mathcal{M}_\iota$ is a Riemannian manifold, $\mathfrak{d}_\iota$ is a Riemannian metric such that $\mathfrak{d}_\iota(u_\iota, v_\iota) = \langle u_\iota, v_\iota\rangle$. Thereby,

$$\mathfrak{d}_{G_l}(u_{G_l}, v_{G_l}) = \langle u_{G_l}, v_{G_l}\rangle = \sum_{\iota \in \mathcal{I}_{G_l}} \langle u_\iota, v_\iota\rangle \sum_{\iota \in \mathcal{I}_{G_l}} \mathfrak{d}_\iota(u_\iota, v_\iota) \tag{6}$$

and we obtain (3). In order to derive (5), we first compute

$$\left\langle \sum_{\iota \in \mathcal{I}_{G_l}} u_\iota, \sum_{\iota \in \mathcal{I}_{G_l}} v_\iota \right\rangle = \sum_{\iota \in \mathcal{I}_{G_l}} \langle u_\iota, v_\iota \rangle. \tag{7}$$

Then, we use the equations for the Lie bracket by

$$\left[ \sum_{\iota \in \mathcal{I}_{G_l}} u_\iota, \sum_{\iota \in \mathcal{I}_{G_l}} v_\iota \right] = \sum_{\iota \in \mathcal{I}_{G_l}} [u_\iota, v_\iota]. \tag{8}$$

Next, we employ the Koszul's formula [7] by

$$2 \langle \bar{\nabla}_{u_\iota} v_\iota, x_\iota \rangle = u_\iota \langle v_\iota, x_\iota \rangle + v_\iota \langle x_\iota, u_\iota \rangle - x_\iota \langle u_\iota, v_\iota \rangle + \langle x_\iota, [u_\iota, v_\iota] \rangle - \langle v_\iota, [u_\iota, x_\iota] \rangle - \langle u_\iota, [v_\iota, x_\iota] \rangle$$

such that

$$\bar{\nabla}_{\bar{u}}(\bar{v}) = \sum_{\iota \in \mathcal{I}_{G_l}} \bar{\nabla}_{u_\iota}(v_\iota), \tag{9}$$

where $\bar{u} = \sum_{\iota \in \mathcal{I}_{G_l}} u_\iota$ and $\bar{v} = \sum_{\iota \in \mathcal{I}_{G_l}} v_\iota$. Using (4) and definition of the curvature with (6), (7), (8), and (9), we obtain (5).

In order to show that $\mathbb{M}_{G_l}$ has never strictly positive sectional curvature $\mathfrak{c}_{G_l}$ in the metric (3), it is sufficient to show that some sectional curvatures always vanish. Suppose that $U$ is a vector field on $\mathbb{M}_{G_l}$ along a component weight manifold $\mathcal{M}_\iota$ such that no local coordinate $o$ of $\mathcal{M}_{\bar{\iota}}$ and $\frac{\partial}{\partial o}$ are present in local coordinates of $U$, $\forall \iota \neq \bar{\iota}$, $\bar{\iota} \in \mathcal{I}_{G_l}$. In addition, suppose that $\bar{U}$ is a vector field along $\mathcal{M}_{\bar{\iota}}$. Then, $\bar{\nabla}_U \bar{U} = 0$, $\forall \iota, \bar{\iota} \in \mathcal{I}_{G_l}$. By employing (9), we have $\bar{C}_\iota(u_\iota, v_\iota, x_\iota, y_\iota) = 0$. Then, we use (5) to obtain $\bar{C}_{G_l}(u_{G_l}, v_{G_l}, x_{G_l}, y_{G_l}) = 0$. Therefore, following the definition of the sectional curvature, for arbitrary vector fields on component manifolds, $\mathbb{M}_{G_l}$ has never strictly positive sectional curvature $\mathfrak{c}_{G_l}$ in the metric (3). Since $\mathbb{M}_{G_l}$ is a Riemannian manifold, if $\mathbb{M}_{G_l}$ is compact, then $\mathbb{M}_{G_l}$ does not admit a metric with negative sectional curvature $\mathfrak{c}_{G_l}$ by the Preissmann's theorem [8] [1].

$\square$

***Theorem* 1** (Computation of gradients on tangent spaces). The $\ell_2$ norm $\|\mathrm{grad}\mathcal{L}(\omega_{G_l^m}^t)\|_2$ of the gradient $\mathrm{grad}\mathcal{L}(\omega_{G_l^m}^t)$ residing on $\mathcal{T}_{\omega_{G_l^m}^t}\mathbb{M}_{G_l^m}$ at the $t^{th}$ epoch and the $l^{th}$ layer can be computed by

$$\|\mathrm{grad}\mathcal{L}(\omega_{G_l^m}^t)\|_2 = \left( \sum_{\iota \in \mathcal{I}_{G_l^m}} \mathrm{grad}\mathcal{L}(\omega_{l,\iota}^t)^2 \right)^{\frac{1}{2}}, \tag{10}$$

where $\mathrm{grad}\mathcal{L}(\omega_{l,\iota}^t)$ is the gradient computed for the weight $\omega_{l,\iota}^t$ on the tangent space $\mathcal{T}_{\omega_{\iota,l}^t}\mathbb{M}_\iota$, $\forall \iota \in \mathcal{I}_{G_l^m}$. ∎

*Proof.* We use the inner product for the Riemannian metric $\mathfrak{d}_{G_l}(\mathrm{grad}\mathcal{L}(\omega_{G_l^m}^t), \mathrm{grad}\mathcal{L}(\omega_{G_l^m}^t))$ and $\mathfrak{d}_\iota(\mathrm{grad}\mathcal{L}(\omega_{l,\iota}^t), \mathrm{grad}\mathcal{L}(\omega_{l,\iota}^t))$ of manifolds $\mathbb{M}_{G_l^m}$ and $\mathbb{M}_\iota$, $\forall \iota$, respectively. By definition of the product manifold, we have

$$\mathrm{grad}\mathcal{L}(\omega_{G_l^m}^t) = \left( \mathrm{grad}\mathcal{L}(\omega_{l,1}^t), \mathrm{grad}\mathcal{L}(\omega_{l,2}^t), \mathrm{grad}\mathcal{L}(\omega_{l,|\mathcal{I}_{G_l}|}^t) \right). \tag{11}$$

Thereby, we can apply bilinearity of inner product in Lemma 1 and obtain

$$\|\mathrm{grad}\mathcal{L}(\omega_{G_l^m}^t)\|_2^2 = \left( \sum_{\iota \in \mathcal{I}_{G_l^m}} \mathrm{grad}\mathcal{L}(\omega_{l,\iota}^t)^2 \right), \tag{12}$$

where $\| \cdot \|_2^2$ is the squared $\ell_2$ norm. The result follows by applying the square root to (12). $\square$

***Theorem* 2** (Convergence of the FG-SGD). Suppose that there exists a local minimum $\hat{\omega}_{G_l} \in \mathbb{M}_{G_l}$, $\forall G_l \subseteq \mathcal{G}_l$, $\forall l$, and $\exists \epsilon > 0$ such that $\inf_{\rho_{G_l}^t > \epsilon^{\frac{1}{2}}} \left\langle \phi_{\omega_{G_l}^t}(\hat{\omega}_{G_l})^{-1}, \nabla \mathcal{L}(\omega_{G_l}^t) \right\rangle < 0$, where $\phi$ is an exponential map or a twice continuously differentiable retraction, and $\langle \cdot, \cdot \rangle$ is the inner product. Then, the loss function and the gradient converges almost surely (a.s.) by $\mathcal{L}(\omega_{G_l}^t) \xrightarrow[t \to \infty]{\text{a.s.}} \mathcal{L}(\hat{\omega}_{G_l})$, and $\nabla \mathcal{L}(\omega_{G_l}^t) \xrightarrow[t \to \infty]{\text{a.s.}} 0$, for each $\mathbb{M}_{G_l}$, $\forall l$. ∎

*Proof.* In this theorem, we generalize the proof idea of Theorem 4.1 and 4.2 given in [9], and Theorem 3 given in [10] for collections of products of embedded weight manifolds (POMs) for training of CNNs. The proof idea is to show that $\rho_{G_l}^t \triangleq \rho(\omega_{l,\iota}^t, \hat{\omega}_{l,\iota})$ converges almost surely to $0$ as $t \to \infty$. For this purpose, we need to first model the change of gradient on the geodesic $\rho_{G_l}^t$ by defining a function $\Psi_t \triangleq \psi((\rho_{G_l}^t)^2)$ according to the following constraints [10];

- $\Psi_t = 0$, for $0 \leq \rho_{G_l}^t \leq \sqrt{\epsilon}$.

- $0 < \Psi_t'' \leq 2$, for $\sqrt{\epsilon} \leq \rho_{G_l}^t \leq \sqrt{\epsilon + 1}$.

- $\Psi_t' = 1$, for $\rho_{G_l}^t \geq \sqrt{\epsilon + 1}$.

Then, we compute gradients and geodesics on collections of POMs using (3) given in Lemma **1** by

$$\|\mathrm{grad}\mathcal{L}(\omega_{G_l}^t)\|_2 = \Big( \sum_{\omega_{l,\iota}^t \in \mathbb{M}_\iota, \iota \in \mathcal{I}_{G_l}} \mathrm{grad}\mathcal{L}(\omega_{l,\iota}^t)^2 \Big)^{\frac{1}{2}} \tag{13}$$

and

$$\rho(\omega_{G_l}^t) = \Big( \sum_{\omega_{l,\iota}^t \in \mathbb{M}_\iota, \iota \in \mathcal{I}_{G_l}} \rho(\omega_{l,\iota}^t, \hat{\omega}_{l,\iota}) \Big), \tag{14}$$

where $\omega_{G_l}^t = (\omega_1^t, \omega_2^t, \cdots, \omega_{|\mathcal{I}_{G_l}|}^t)$. We employ a Taylor expansion on $\Psi_t$ [10, 9], and we obtain

$$\Psi_{t+1} - \Psi_t \leq ((\rho_{G_l}^{t+1})^2 - (\rho_{G_l}^t)^2)\Psi_t' + ((\rho_{G_l}^{t+1})^2 - (\rho_{G_l}^t)^2)^2. \tag{15}$$

In order to compute the difference between $\rho_{G_l}^{t+1}$ and $\rho_{G_l}^t$, we employ a Taylor expansion on the geodesics [10, 9] by

$$\rho_{G_l}^{t+1} - \rho_{G_l}^t \leq \Big( \frac{g(t, \Theta)}{\mathfrak{g}(\omega_{G_l}^t)} \Big)^2 \|\mathrm{grad}\mathcal{L}(\omega_{G_l}^t)\|^2 \kappa - 2 \Big\langle h(\mathrm{grad}\mathcal{L}(\omega_{G_l}^t), g(t, \Theta)), \phi_{\omega_{G_l}^t}(\hat{\omega}_{G_l})^{-1} \Big\rangle,$$

where $\hat{\omega}_{G_l} = (\hat{\omega}_1, \hat{\omega}_2, \cdots, \hat{\omega}_{|\mathcal{I}_{G_l}|})$, and $\kappa \leq \Upsilon_1$ where $\Upsilon_1 = 1 + \mathfrak{c}_{G_l}(\rho_{G_l}^t + R_{G_l}^t)$ is an upper bound on the operator norm of half of the Riemannian Hessian of $\rho(\cdot, \hat{\omega}_{G_l})^2$ along the geodesic joining $\omega_{G_l}^t$ and $\omega_{G_l}^{t+1}$. In order to explore asymptotic convergence, we define $\Omega_t = \{s_i\}_{i=1}^{t-1}$ to be an increasing sequence of $\sigma$ algebras generated by samples that are processed before the $t^{th}$ epoch. Since $s_t$ is independent of $\Omega_t$ and $\omega_{G_l}^t$ is $\Omega_t$ measurable, we have

$$\mathbb{E}(h(\mathrm{grad}\mathcal{L}(\omega_{G_l}^t), g(t, \Theta))^2 \kappa | \Omega_t]) \leq \Big( \frac{g(t, \Theta)}{\mathfrak{g}(\omega_{G_l}^t)} \Big)^2 \mathbb{E}\Big( (R_{G_l}^t)^2 \Upsilon_1 \Big), \tag{16}$$

and

$$\mathbb{E}((\rho_{G_l}^{t+1})^2 - (\rho_{G_l}^t)^2 | \Omega_t) \leq 2 \frac{g(t, \Theta)}{\mathfrak{g}(\omega_{G_l}^t)} \Big\langle \phi_{\omega_{G_l}^t}(\hat{\omega}_{G_l})^{-1}, \nabla\mathcal{L}(\omega_{G_l}^t) \Big\rangle + g(t, \Theta)^2. \tag{17}$$

If $\mathfrak{g}(\omega_{G_l}^t) = \max\{1, \Gamma_1^t\}^{\frac{1}{2}}$, $\Gamma_1^t = (R_{G_l}^t)^2 \Gamma_2^t$, $\Gamma_2^t = \max\{(2\rho_{G_l}^t + R_{G_l}^t)^2, (1 + \mathfrak{c}_{G_l}(\rho_{G_l}^t + R_{G_l}^t))\}$, then we have

$$\mathbb{E}(\Psi_{t+1} - \Psi_t | \Omega_t) \leq \mathbb{E}((\rho_{G_l}^{t+1})^2 - (\rho_{G_l}^t)^2 | \Omega_t)\Psi_t' + g(t, \Theta)^2 \tag{18}$$

and

$$\mathbb{E}(\Psi_{t+1} - \Psi_t | \Omega_t) \leq 2 \frac{g(t, \Theta)}{\mathfrak{g}(\omega_{G_l}^t)} \Big\langle \phi_{\omega_{G_l}^t}(\hat{\omega}_{G_l})^{-1}, \nabla\mathcal{L}(\omega_{G_l}^t) \Big\rangle \Psi_t' + g(t, \Theta)^2. \tag{19}$$

Thus, we have

$$\mathbb{E}(\Psi_{t+1} - \Psi_t | \Omega_t) \leq 2g(t, \Theta)^2, \tag{20}$$

and $\Psi_t + \sum_{t=0}^{\infty} g(t, \Theta)^2$ is a positive supermartingale, and converges almost surely. Since

$$\sum_{t=0}^{\infty} \mathbb{E}([\mathbb{E}(\Psi_{t+1} - \Psi_t | \Omega_t)^+]) \leq \sum_{t=0}^{\infty} g(t, \Theta)^2 < \infty, \tag{21}$$

we observe that $\Psi_t$ is a quasi-martingale [10, 9], and thereby we have almost surely

$$-\sum_{t=0}^{\infty} \frac{g(t,\Theta)}{\mathfrak{g}(\omega_{G_l}^t)} \left\langle \phi_{\omega_{G_l}^t}(\hat{\omega}_{G_l})^{-1}, \nabla\mathcal{L}(\omega_{G_l}^t) \right\rangle \Psi_t' < \infty. \tag{22}$$

Using properties of quasi-martingale [11], $\Psi_t$ converges almost surely. In order to show almost sure convergence of $\nabla\mathcal{L}(\omega_{G_l}^t)$ to 0, we use Theorem 4.1 and 4.2 of [9]. For this purpose, we need to show that gradients of loss functions are bounded in compact sets of weights. Since $\inf_{\rho_{G_l}^t > \epsilon^{\frac{1}{2}}} \left\langle \phi_{\omega_{G_l}^t}(\hat{\omega}_{G_l})^{-1}, \nabla\mathcal{L}(\omega_{G_l}^t) \right\rangle < 0$, a weight $\omega_{G_l}^t$ is moved towards $\hat{\omega}_{G_l}$ by the gradient when $\rho_{G_l}^t > \epsilon^{\frac{1}{2}}$ where the set $\mathfrak{S} = \{\omega_{G_l}^t : \rho_{G_l}^t \leq \epsilon^{\frac{1}{2}}\}$ is a compact set. Since all continuous functions of $\omega_{G_l}^t \in \mathfrak{S}$ are bounded, and adaptive step size $\mathfrak{g}(\omega_{G_l}^t)$ satisfies $\frac{g(t,\Theta)}{\mathfrak{g}(\omega_{G_l}^t)} \leq g(t,\Theta)$ and $\mathfrak{g}(\omega_{G_l}^t)^2$ dominates $R_{G_l}^t$, we obtain that $\mathbb{E}(R_{G_l}^t)^2 \leq \mathfrak{K}$ for some $\mathfrak{K} > 0$ on a compact set $\mathcal{K}$. Thereby, we can show that conditions of Theorem 4.1 and 4.2 of [9] are satisfied. Therefore, we obtain almost sure convergence of $\nabla\mathcal{L}(\omega_{G_l}^t)$ to 0 by applying Theorem 4.1 and 4.2 in the rest of the proof.

□

***Corollary* 1**. Suppose a DNN has loss functions whose local minima are also global minima. If the DNN is trained using the proposed FG-SGD and weight renormalization methods, then the loss of the DNN converges to global minima.

*Proof.* By Theorem 2, we assure that a loss function of a DNN which employs the proposed FG-SGD and weight renormalization methods for training converges to local minima. If the local minima is the global minima for the DNN, then the loss function converges to the global minima. □

***Corollary* 2**. Suppose that $\mathbb{M}_\iota$ are identified by $n_\iota \geq 2$ dimensional unit sphere $\mathbb{S}^{n_\iota}$, and $\rho_{G_l}^t \leq \hat{\mathfrak{c}}^{-1}$, where $\hat{\mathfrak{c}}$ is an upper bound on the sectional curvatures of $\mathbb{M}_{G_l}, \forall l$ at $\omega_{G_l}^t \in \mathbb{M}_{G_l}, \forall t$. If step size is computed using

$$h(\mathrm{grad}\mathcal{L}(\omega_{G_l}^t), g(t,\Theta)) = -\frac{g(t,\Theta)}{\mathfrak{g}(\omega_{G_l}^t)}\mathrm{grad}\mathcal{L}(\omega_{G_l}^t), \tag{23}$$

with $\quad \mathfrak{g}(\omega_{G_l}^t) = (\max\{1, (R_{G_l}^t)^2(2+R_{G_l}^t)^2\})^{\frac{1}{2}}, \quad$ then $\quad \mathcal{L}(\omega_{G_l}^t) \xrightarrow[t\to\infty]{a.s.} \mathcal{L}(\hat{\omega}_{G_l}), \quad$ and $\nabla\mathcal{L}(\omega_{G_l}^t) \xrightarrow[t\to\infty]{a.s.} 0$, for each $\mathbb{M}_{G_l}, \forall l$. ∎

*Proof.* If $\mathbb{M}_{G_l}$ is a product of $n_\iota \geq 2$ dimensional unit spheres $\mathbb{S}^{n_\iota}$, then $\mathfrak{c}_{G_l} = 0$ and $\hat{\mathfrak{c}} = 1$ by Lemma **1**. Thereby, Theorem **2** is applied to assure convergence by $\Gamma_t^1 = (R_{G_l}^t)^2(2+R_{G_l}^t)^2$.

□

# 3   Experimental Details

We use three benchmark image classification datasets, namely Cifar-10, Cifar-100 and Imagenet [12], for analysis of convergence properties and performance of CNNs trained using FG-SGD. The Cifar-10 dataset consists of 60000 $32 \times 32$ RGB images (50000 training images and 10000 test images) in 10 classes, with 6000 images per class. The Cifar-100 dataset consists of 100 classes containing 600 images each (500 training images and 100 testing images per class). The Imagenet (ILSVRC 2012) dataset consists of 1000 classes of $224 \times 224$ RGB images (1.2 million training images, 100000 test images and 50000 images used for validation).

## 3.1   Computational Complexity of Algorithm 1

Compared to SGD algorithms that use weights belonging to linear weight spaces [13, 14], the computational complexity of Algorithm 1 is dominated by computation of the maps $\Pi$ and $\phi$ at line 6 and 9, depending on the structure of the weight manifold used at the $l^{th}$ layer. Concisely, the computational complexity of $\Pi$ is determined by computation of different norms that identify the manifolds. For instance, for the sphere, we use $\Pi_{\omega_l^t}\mu_t \triangleq (1 - \|\omega_l^t\|_F^2)\mu_t$. Thereby, for an $A \times A$

weight, the complexity is bounded by $O(A^3)$, where $O(\cdot)$ denotes an asymptotic upper bound [15]. Similarly, the computational complexity of $\phi$ depends on the manifold structure. For example, the exponential maps on the sphere and the oblique manifold can be computed using functions of $\sin$ and $\cos$ functions, while that on the Stiefel manifold is a function of matrix exponential. For computation of matrix exponential, various numerical approximations with $O(\epsilon A^3)$ complexity were proposed for different approximation order $\epsilon$ [11, 16, 17, 18]. However, unit norm matrix normalization is used for computation of retractions on the sphere and the oblique manifold. Moreover, QR decomposition of matrices is computed with $O(A^3)$ [19] for retractions on the Stiefel manifold. In addition, computation time of maps can be reduced using parallel computation methods. For instance, a rotation method was suggested to compute QR using $O(A^2)$ processors in $O(A)$ unit time in [20]. Therefore, computation of retractions is computationally less complex compared to that of the exponential maps. Since the complexity analysis of these maps is beyond the scope of this work, and they provide the same convergence properties for our proposed algorithm, we used the retractions in the experiments. Implementation details are given in the next section.

### 3.1.1 A Discussion on Implementation of Algorithm 1 in Parallel and Distributed Computing Systems

In the experiments, algorithms are implemented using GPU and CPU servers consisting of GTX 2070, GTX 1080, GTX-Titan-X, GTX-Titan-Black, Intel i7-5930K, Intel Xeon E5-1650 v3 and E5-2697 v2. Since we used hybrid GPU and CPU servers in the experiments, and a detailed analysis of parallel and distributed computation methods of CNNs is beyond the scope of this work, we report bounds on average running times of SGD algorithms in this section.

In the implementation of linear Euclidean SGD methods, we use vectorized computation of weight updates. Therefore, we use large scale matrix computation methods (in some cases, for sparse matrices) to improve running time of the linear Euclidean SGD methods. However, we deal with optimization using batched (small size) dense matrices in the implementation of Algorithm 1 [21]. Therefore, in order to improve running time of the algorithm, we implemented Algorithm 1 using hybrid CPU-GPU programming paradigms.

More precisely, we consider two computation schemes according to matrix/tensor structure of the weights, i.e. geometric structure of weight manifolds. First, we recall that we construct different manifolds of weights $\mathcal{W} = \left\{ \mathbf{W}_{d,l} \in \mathbb{R}^{A_l \times B_l \times C_l} \right\}_{d=1}^{D_l}, \forall l = 1, 2, \ldots, L$, at different layers of an $L$-layer CNN. Then, we implement projections of gradients and retractions at

1. Fully Connected (FC) layers at which we use $\mathbf{W}_l^{fc} \in \mathbb{R}^{C_l \times D_l}$ with $A_l = B_l = 1$, and

2. Convolution (Conv) layers at which we use $\mathbf{W}_{d,l} \in \mathcal{W}$ with $A_l > 1$ and $B_l > 1$.

At the FC layers, we implemented Algorithm 1 on GPUs using Cuda with Cublas and Magma [22, 23, 24] Blas [25, 26]. In the experimental analyses, we obtained similar running times using Cublas and Magma Blas implementation of Algorithm 1 (denoted by $\mathcal{R}_M^{fc}$) compared to running time of linear Euclidean SGD (denoted by $\mathcal{R}_E^{fc}$), for each epoch.

For instance, if we train CNNs using the Cifar-100 dataset and one GTX 1080, then we observe $\mathcal{R}_M^{fc} < \mathfrak{a} \mathcal{R}_E^{fc}$, where the running times are bounded by $\mathfrak{a} > 0$ due to implementation of gradient projections and retractions. The overhead factor $\mathfrak{a}$ also depends on the manifold structure of the weights such that $\mathfrak{a} < 1.5$ for the sphere, $\mathfrak{a} < 2.5$ for the oblique manifold and $\mathfrak{a} < 5$ for the Stiefel manifold.

When we implemented a QR decomposition algorithm using the Givens transformation (Rotation) [27, 19], we obtained further improvement by $\mathfrak{a} < 4$. In addition, batch size does not affect the overhead of running time crucially as long as the GPU memory is sufficient. The effect of this overhead on the overall training time depends on structure of CNNs. For example, we use multiple (6) FC layers in NiNs where we have 2 FC layers in SKs. Therefore, the overhead affects the training time of NiNs more than that of SKs.

At the Conv layers, we implemented Algorithm 1 on both GPUs and CPUs. However, the structure of parallelization of projections and maps at the Conv layers is different than that of projections and maps computed at the FC layers. More precisely, we perform parallel computation either 1) using

tensors $\mathbf{W}_{d,l} \in \mathbb{R}^{A_l \times B_l \times C_l}$ for each output $d = 1, 2, \ldots, D_l$, or 2) using matrices $W_{c,d,l} \in \mathbb{R}^{A_l \times B_l}$ for each output $d = 1, 2, \ldots, D_l$ and channel $c = 1, 2, \ldots, C_l$.

Since there is an I/O bottleneck between transfer of matrices and tensors to/from GPUs from/to CPUs, we used either (1) or (2) according to output size $D_l$, and channel size $C_l$. For instance, if $C_l > D_l$, then we performed computations on GPUs. Otherwise, we implemented the algorithm on multi-core CPUs.

In average, for an epoch[2], the running time of a GPU implementation of Algorithm 1 for the case (1) denoted by $\mathcal{R}^1_{M,gpu}$, and that of linear Euclidean SGD for the case $\mathcal{R}^1_{E,gpu}$ are related by $\mathcal{R}^1_{E,gpu} < \mathfrak{a}\mathcal{R}^1_{M,gpu}$ for $\mathfrak{a} < 3$ for the sphere and $\mathfrak{a} < 3$ for the oblique manifold and $\mathfrak{a} < 6$ for the Stiefel manifold[3]. The additional computational overhead can be attributed to additional transmission time and computation of multi-dimensional transpose operations.

Moreover, we observed that the running time of the multi-core CPU implementation of the algorithm $\mathcal{R}^1_{M,cpu}$ is bounded by $\mathcal{R}^1_{M,gpu} < \mathfrak{a}\mathcal{R}^1_{M,cpu}$ for $\mathfrak{a} < f(D_l) < 10$, where $f(\cdot)$ is a function of number of output $D_l$ for all manifolds[4]. In other words, the difference between running times on CPUs and GPUs is affected by $D_l$ more than the other parameters $2 \le A_l \le 7$ and $2 \le B_l \le 7$, and $C_l$. This observation can be attributed to the less overhead between Blas and Cublas implementations of matrix operations for small number (e.g. $C_l < 10^3$) of weight matrices.

For the second case where $C_l > D_l$, we observed that $\mathcal{R}^1_{E,gpu} < \mathfrak{a}_1 \mathcal{R}^1_{M,cpu} < \mathfrak{a}_2 \mathcal{R}^1_{M,gpu}$. We observed that $\mathfrak{a}_1 < \hat{f}(C_l, D_l) < 2$ and $\mathfrak{a}_2 < \hat{f}(C_l, D_l) < 5$, where $\hat{f}(\cdot, \cdot)$ is a function of both $C_l$ and $D_l$, for the sphere, and scales for the other manifolds accordingly, for implementation using one GTX 1080 and E5-2697 v2.

## 3.2 Implementation Details of Algorithm 1

In this section, we give implementation details of Algorithm 1.

### 3.2.1 Identification of Component Kernel Submanifolds of POMs

We identify component weight manifolds $\mathcal{M}_\iota$ of POMs $\mathbb{M}_{G_l}$ at each $l^{th}$ of an $L$-layer CNN, and initialize weights residing in the manifolds considering both statistical properties of data, and geometric properties of weight manifolds.

In the experiments, we used the sphere, the oblique manifold and the Stiefel manifold to construct component weight manifolds according to definition of manifolds given in Table 2.

Table 4: Tangent spaces and maps used for orthogonal projection of Euclidean gradients obtained using backpropagation onto the tangent spaces for the manifolds of the normalized weights defined in Table 2. We denote a vector realized by a Euclidean gradient obtained at a weight $\omega^t_{G_l}$ from the $l + 1^{st}$ layer using backpropagation by $\mu \triangleq \left(\mathrm{grad}_E \mathcal{L}(\omega^t_{g,l}), \Theta, \mathcal{R}^t_l\right)$ (see Line 5 of Algorithm 1).

| Manifolds | Tangent Spaces | Projection of Gradients |
|---|---|---|
| $\mathcal{S}(A_l, B_l)$ | $T_\omega \mathcal{S}(A_l, B_l) = \{\hat{\omega} \in \mathbb{R}^{A_l \times B_l} : \omega^{\mathrm{T}}\hat{\omega} = 0\}$ | $\Pi_\omega \mu = (I - \omega\omega^{\mathrm{T}})\mu$ |
| $\mathcal{OB}(A_l, B_l)$ | $T_\omega \mathcal{OB}(A_l, B_l) = \{\hat{\omega} \in \mathbb{R}^{A_l \times B_l} : \omega^{\mathrm{T}}\hat{\omega} = 0\}$ | $\Pi_\omega \mu = \mu - \omega\mathrm{ddiag}(\omega^{\mathrm{T}}\mu)$ |
| $St(A_l, B_l)$ | $T_\omega St(A_l, B_l) = \{\hat{\omega} \in \mathbb{R}^{A_l \times B_l} : \mathrm{ddiag}(\omega^{\mathrm{T}}\hat{\omega}) = 0\}$ | $\Pi_\omega \mu = (I - \omega\omega^{\mathrm{T}})\mu + \omega\varsigma(\omega^{\mathrm{T}}\mu)$ |

Table 5: Exponential maps and retractions for the manifolds of the normalized weights defined in Table 2. We denote a vector moved on a tangent space at the $t^{th}$ epoch by $v_t$ (see Line 8 of Algorithm 1). In addition, $\aleph(Z)$ is the unit-norm normalization of each column of a matrix $Z$. $\mathcal{Q}_{\mathcal{F}}(Z) := Q$ is the $Q$ factor of the QR decomposition $Z = QR$ of $Z$.

| Manifolds | Exponential Maps | Retraction |
|---|---|---|
| $\mathcal{S}(A_l, B_l)$ | $\exp_\omega(v) = \omega\cos(\|v\|_F) + \frac{v}{\|v\|_F}\sin(\|v\|_F)$ | $\mathfrak{R}_\omega(v) = \frac{\omega+v}{\|\omega+v\|_F}$ |
| $\mathcal{OB}(A_l, B_l)$ | $\exp_\omega(v) = \omega\,\mathrm{ddiag}(\cos(\|v\|_F)) + v\,\mathrm{ddiag}(\frac{\sin(\|v\|_F)}{\|v\|_F})$ | $\mathfrak{R}_\omega(v) = \aleph(\omega + v)$ |
| $St(A_l, B_l)$ | $\exp_\omega(v) = [\omega\ v]\hat{\mathrm{exp}}\left(\begin{bmatrix} \omega^{\mathrm{T}}v & -v^{\mathrm{T}}v \\ I & \omega^{\mathrm{T}}v \end{bmatrix}\right)\begin{bmatrix} I \\ 0 \end{bmatrix}\hat{\mathrm{exp}}(-\omega^{\mathrm{T}}v)$ | $\mathfrak{R}_\omega(v) = \mathcal{Q}_{\mathcal{F}}(\omega + v)$ |

### 3.2.2 Computation of Gradient Maps, Projections and Retractions used in Algorithm 1

In this section, we provide the details of the methods used for computation of gradient maps, projections and retractions for different collections of POMs in Algorithm 1. We denote a vector moved on a tangent space at the $t^{th}$ epoch by $v_t$ (see Line 7 of Algorithm 1). In addition, $\aleph(Z)$ is the unit-norm normalization of each column of a matrix $Z$. $\mathcal{Q}_{\mathcal{F}}(Z) := Q$ is the $Q$ factor of the QR decomposition $Z = QR$ of $Z$.

Definitions of component manifolds of POMs used in this work are given in Table 2. In Table 4, we provide tangent spaces and maps used for orthogonal projection of Euclidean gradients onto the tangent spaces for the manifolds of the normalized weights which are defined in Table 2. Exponential maps and retractions are given in Table 5.

We also note that various types of projections, exponential maps and retractions can be computed and used in Algorithm 1 in addition to the projections, maps and retractions given in the tables. More detailed discussion on their computation are given in [28, 29, 6].

### 3.3 Implementation Details of CNN Architectures used in the Experiments

**Data pre-processing and post-processing:** For the experiments on Cifar-10 and Cifar-100 datasets, we used two standard data augmentation techniques which are horizontal flipping and translation by 4 pixels [13, 30].

For the experiments on Imagenet dataset, we followed the data augmentation methods suggested in [13]. In addition, we used both the scale and aspect ratio augmentation used in [31]. For color augmentation, we used the photometric distortions [32] and standard color augmentation [13]. Moreover, we used random sampling of $224 \times 224$ crops or their horizontal flips with the normalized data obtained by subtracting per-pixel mean. In the bottleneck blocks, stride 2 is used for the $A_l = B_l = 3$ weights. Moreover, Euclidean gradient decays are employed for all the weights.

**Acceleration methods:** In this section, we employed state-of-the-art acceleration methods [33] modularly in Algorithm 1 for implementation of the CNNs as suggested in the reference works [13, 30, 9]. In this work, we consider employment of acceleration methods on the ambient Euclidean space and collections of POMs as suggested in [9]. For this purpose, momentum and Euclidean gradient decay methods are employed on the Euclidean gradient $\mathrm{grad}_E\ \mathcal{L}(\omega_{g,l}^t)$ using $\mu_t := q\Big(\mathrm{grad}_E\ \mathcal{L}(\omega_{g,l}^t), \mu_t, \Theta\Big)$. We can employ state-of-the-art acceleration methods [33] modularly in this step. Thus, momentum was employed with the Euclidean gradient decay using

$$q\Big(\mathrm{grad}_E\ \mathcal{L}(\omega_{g,l}^t), \mu_t, \Theta\Big) = \theta_\mu \mu_t - \theta_E \mathrm{grad}_E\ \mathcal{L}(\omega_{g,l}^t), \tag{24}$$

where $\theta_\mu \in \Theta$ is the parameter employed on the momentum variable $\mu_t$. We consider $\theta_E \in \Theta$ as the decay parameter for the Euclidean gradient. In the experiments, we used $\theta_\mu = \theta_E = 0.9$.

**Architectural Details of CNNs:** In the experiments, we used the same hyper-parameters of CNN architectures (e.g. number of channels, layers, weight sizes, stride and padding parameters) and their implementation provided by the authors of the compared works for training of CNNs using our proposed SGD method, for a fair comparison with base-line methods. Differences between the implementations and hyper-parameters are explained below. In other words, we just implemented the SGD algorithm of the provided CNN implementations using our proposed SGD method. More precisely, we used the following implementations for comparison:

- RCD and RSD: We used the Residual networks with constant and stochastic depth using the same configuration hyper-parameters (see below for number of weights used in the architectures) and code given in [30].

- Residual Networks (Resnets): We re-implemented residual networks with the same configuration and training hyper-parameters (see below for number of weights used in the architectures) given in [13, 9].

- Squeeze-and-Excitation networks implemented for Resnets with 50 layers (SENet-Resnet-50): We re-implemented residual networks with the same configuration and training hyper-parameters (see below for number of weights used in the architectures) given in [34].

In order to construct collections of weights belonging to four spaces (Euc., Sp, St and Ob) using WSS, we increase the number of weights used in CNNs to 24 and its multiples as follows;

• Resnet with 18 Layers (Table 6 in this text): 72 filters at the first and second, 144 filters at the third, 288 filters at the fourth, and 576 filters at the fifth convolution blocks [13].

• Resnet with 44 Layers (Table 7 in this text): 24 filters for 15 layers, 48 filters for 14 layers, 96 filters for 14 [13].

• Resnets with constant depth (RCD) and stochastic depth (RSD) with 110 layers (Table 2 in the main text and Table 8 in this text): 24, 48 and 72 filters at the first, second, and the third convolution blocks [30].

• Resnet-50 and SENet-Resnet-50 (Table 1 in the main text): Configurations of Resnet-50 and SENet-Resnet-50 are given in Table 6 and Table 7, respectively.

**Scaling of weights:** We use $\mathfrak{R}_l^t$ for scaling of weights and identification of component weight manifolds of POMs. As we mentioned in the main text, for instance, $\mathfrak{R}_l^t$ is computed and used as the radius of the sphere. More precisely, we initialize weights $\omega \in \mathcal{M}_\iota$ that belong to the sphere $\mathcal{M}_\iota \equiv \mathcal{S}(A_l, B_l)$ subject to the constraint $\|\omega\|_F^2 = \mathfrak{R}_l^t$ by constructing a scaled sphere

$$\mathbb{S}^{A_l B_l - 1} \triangleq \mathcal{S}_{\mathfrak{R}_l^t}(A_l, B_l) = \{\omega \in \mathbb{R}^{A_l \times B_l} : \|\omega\|_F^2 = \mathfrak{R}_l^t\}. \tag{25}$$

The other manifolds (the oblique and the Stiefel manifolds) are identified, and the weights that belong to the manifolds are initialized, appropriately, following the aforementioned methods. Then, projection of gradients, exponential maps and retractions which are determined according to manifold structure of weight spaces (see Table 4 and Table 5), are updated accordingly by $\mathfrak{R}_l^t$. For example, for the scaled sphere $\mathcal{S}_{\Gamma_l^t}(A_l, B_l)$, we compute the projection of gradients by $(I\mathfrak{R}_l^t - \omega\omega^T)\mu$, and the exponential map by

$$\exp_\omega(v) = \omega \cos(\|v\|_F \mathfrak{R}_l^t) + \mathfrak{R}_l^t \frac{v}{\|v\|_F} \sin(\|v\|_F \mathfrak{R}_l^t). \tag{26}$$

Table 6: Configuration details of the Resnet-50 used for the experiments given in Table 1 in the main text.

| Output Size | Resnet-50 |
|---|---|
| $112 \times 112$ | Kernel size: $7 \times 7$, Number of convolution weights: 64, Stride 2 |
| $56 \times 56$ | $3 \times 3$ Max Pooling, Stride 2<br>3 Residual Blocks with the Following Convolution Kernels:<br>72 convolution weights of size $1 \times 1$<br>72 convolution weights of size $3 \times 3$<br>264 convolution weights of size $1 \times 1$ |
| $28 \times 28$ | 4 Residual Blocks with the Following Convolution Kernels:<br>144 convolution weights of size $1 \times 1$<br>144 convolution weights of size $3 \times 3$<br>528 convolution weights of size $1 \times 1$ |
| $14 \times 14$ | 6 Residual Blocks with the Following Convolution Kernels:<br>264 convolution weights of size $1 \times 1$<br>264 convolution weights of size $3 \times 3$<br>1032 convolution weights of size $1 \times 1$ |
| $7 \times 7$ | 3 Residual Blocks with the Following Convolution Kernels:<br>528 convolution weights of size $1 \times 1$<br>528 convolution weights of size $3 \times 3$<br>2064 convolution weights of size $1 \times 1$ |
| $1 \times 1$ | Global Average Pooling<br>Fully connected layer<br>Softmax |

### 3.4 Employment of Weight Set Splitting Scheme (WSS) in the Experiments:

Recall that, at each $l^{th}$ layer, we compute a weight $\omega_\iota \triangleq W_{c,d,l}$, $c \in \Lambda^l$, $\Lambda^l = \{1, 2, \ldots, C_l\}$, $d \in O^l$, $O^l = \{1, 2, \ldots, D_l\}$. We first choose $\mathfrak{A}$ subsets of indices of input channels $\Lambda_a \subseteq \Lambda^l, a = 1, 2, \ldots, \mathfrak{A}$, and $\mathfrak{B}$ subsets of indices of output channels $O_b \subseteq O^l, b = 1, 2, \ldots, \mathfrak{B}$, such that $\Lambda^l = \bigcup_{a=1}^{\mathfrak{A}} \Lambda_a$ and $O^l = \bigcup_{b=1}^{\mathfrak{B}} O_b$. We determine indices of weights belonging to different groups using the following three schemes:

1. POMs for input channels (PI): For each $c^{th}$ input channel, we construct $\mathcal{I}_{\mathcal{G}_l} = \bigcup_{c=1}^{C_l} \mathcal{I}_{G_l}^c$, where $\mathcal{I}_{G_l}^c = O_b \times \{c\}$ and the Cartesian product $O_b \times \{c\}$ preserves the input channel index, $\forall b, c$ (see Figure 1).

2. POMs for output channels (PO): For each $d^{th}$ output channel, we construct $\mathcal{I}_{\mathcal{G}_l} = \bigcup_{d=1}^{D_l} \mathcal{I}_{G_l}^d$, where $\mathcal{I}_{G_l}^d = \Lambda_a \times \{d\}$ and the Cartesian product $\Lambda_a \times \{d\}$ preserves the output channel index, $\forall a, d$ (see Figure 1).

3. POMs for input and output channels (PIO): In PIO, we construct $\mathcal{I}_l^{a,b} = \mathcal{I}_l^a \cup \mathcal{I}_l^b$, where $\mathcal{I}_l^a = \{\Lambda_a \times a\}$, and $\mathcal{I}_l^b = \{O_b \times b\}$ such that $\mathcal{I}_{\mathcal{G}_l} = \bigcup_{a=1,b=1}^{\mathfrak{A},\mathfrak{B}} \mathcal{I}_l^{a,b}$ (see Figure 1).

**Illustrative Examples of Employment of PI, PO and PIO**

A comparative and illustrative example for comparison of PI, PO and PIO is given in Figure 1.

Table 7: Configuration details of the SENet-Resnet-50 used for the experiments given in Table 1 in the main text.

| Output Size | Resnet-50 |
|---|---|
| $112 \times 112$ | Kernel size: $7 \times 7$, Number of convolution weights: 64, Stride 2 |
| $56 \times 56$ | $3 \times 3$ Max Pooling, Stride 2<br>3 Residual Blocks with the Following Convolution Kernels:<br>72 convolution weights of size $1 \times 1$<br>72 convolution weights of size $3 \times 3$<br>264 convolution weights of size $1 \times 1$<br>Fully connected layer with weights of size $24 \times 264$ |
| $28 \times 28$ | 4 Residual Blocks with the Following Convolution Kernels:<br>144 convolution weights of size $1 \times 1$<br>144 convolution weights of size $3 \times 3$<br>528 convolution weights of size $1 \times 1$<br>Fully connected layer with weights of size $48 \times 528$ |
| $14 \times 14$ | 6 Residual Blocks with the Following Convolution Kernels:<br>264 convolution weights of size $1 \times 1$<br>264 convolution weights of size $3 \times 3$<br>1032 convolution weights of size $1 \times 1$<br>Fully connected layer with weights of size $72 \times 1032$ |
| $7 \times 7$ | 3 Residual Blocks with the Following Convolution Kernels:<br>528 convolution weights of size $1 \times 1$<br>528 convolution weights of size $3 \times 3$<br>2064 convolution weights of size $1 \times 1$<br>Fully connected layer with weights of size $144 \times 2064$ |
| $1 \times 1$ | Global Average Pooling<br>Fully connected layer<br>Softmax |

Figure 1: An illustration for employment of the proposed PI, PO and PIO strategies at the $l^{th}$ layer of a CNN.

**Example 3.1.** Suppose that we have a weight tensor of size $3 \times 3 \times 4 \times 6$ where the number of input and output channels is $4$ and $6$. In total, we have $4 * 6 = 24$ weight matrices of size $3 \times 3$. An example of construction of an collection of POMs is as follows.

1. PIO: We split the set of 24 weights into 10 subsets. For 6 output channels, we split the set of weights corresponding to 4 input channels into 3 subsets. We choose the sphere (Sp) for 2 subsets each containing 3 weights (depicted by light blue rectangles), and 3 subsets each containing 2 weights (depicted by red rectangles). We choose the Stiefel manifold (St) similarly for the remaining subsets. Then, our ensemble contains 5 POMs of St and 5 POMs of Sp.

2. PI: For each of 4 input channels, we split a set of 6 weights associated with 6 output channels into two subsets of 3 weights. Choosing the sphere (Sp) for the first subset, we construct a POM

as a product of 3 Sp. That is, each of 3 component manifolds $\mathcal{M}_\iota, \iota = 1, 2, 3$, of the POM is a sphere. Similarly, choosing the Stiefel (St) for the second subset, we construct another POM as a product of 3 St (each of 3 component manifolds $\mathcal{M}_\iota, \iota = 1, 2, 3$, of the second POM is a Stiefel manifold.). Thus, at this layer, we construct an collection of 4 POMs of 3 St and 4 POMs of 3 Sp.

3. PO: For each of 6 output channels, we split a set of 4 weights corresponding to the input channels into two subsets of 2 weights. We choose the Sp for the first subset, and we construct a POM as a product of 2 Sp using. We choose the St for the second subset, and we construct a POM as a product of 2 St. Thereby, we have an collection consisting of 6 POMs of St and 6 POMs of Sp.

In the experiments, indices of weights for PI, PO and PIO are randomly selected. An illustration of the selection method is given in Figure 2.

Figure 2: An illustration of employment of the proposed PI, PO and PIO collection strategies at the $l^{th}$ layer of a CNN. In Section 4.2, we randomly selected indices of weights, i.e. subsets of input and output channels, according to the uniform distribution. In this example, we suppose that there are four input and six output channels. Then, 24 convolution weights are computed on in two different POMs.

Table 8: Results for Resnet-18 which are trained using the Imagenet for single crop validation error rate (%).

| Model | Top-1 Error (%) |
|---|---|
| Euc. [9] | 30.59 |
| Euc. † | 30.31 |
| Sp/Ob/St[9] | 29.13/28.97/28.14 |
| Sp/Ob/St † | 28.71/28.83/28.02 |
| POMs of Sp/Ob/St | 28.70/28.77/28.00 |
| PI for POMs of Sp/Ob/St | 28.69/28.75/27.91 |
| PI (Euc.+Sp/Euc.+St/Euc.+Ob) | 30.05/29.81/29.88 |
| PI (Sp+Ob/Sp+St/Ob+St) | 28.61/28.64/28.49 |
| PI (Sp+Ob+St/Sp+Ob+St+Euc.) | 27.63/27.45 |
| PO for POMs of Sp/Ob/St | 28.67/28.81/27.86 |
| PO (Euc.+Sp/Euc.+St/Euc.+Ob) | 29.58/29.51/29.90 |
| PO (Sp+Ob/Sp+St/Ob+St) | 28.23/28.01/28.17 |
| PO (Sp+Ob+St/Sp+Ob+St+Euc.) | 27.81/27.51 |
| PIO for POMs of Sp/Ob/St | 28.64/28.72/27.83 |
| PIO (Euc.+Sp/Euc.+St/Euc.+Ob) | 29.19/28.25/28.53 |
| PIO (Sp+Ob/Sp+St/Ob+St) | 28.14/27.66/27.90 |
| PIO (Sp+Ob+St/Sp+Ob+St+Euc.) | 27.11/27.07 |

**Notation used in the Tables**

1. Sp/Ob/St: Kernels employed on each input and output channel are defined to reside on the sphere, oblique and Stiefel manifold, respectively.

2. POMs of Sp/Ob/St: Kernels employed on all input and output channels are defined to reside on a POM of Sp/Ob/St.

3. PI/PO/PIO for POMs of Sp/Ob/St: Ensembles of POMs of Sp/Ob/St are computed using the schemes PI/PO/PIO.

4. Results for Manifold$_1$ + Manifold$_2$: Results are computed for collections of POMs of Manifold$_1$ and Manifold$_2$.

5. Results for Manifold$_1$ + Manifold$_2$ + Manifold$_3$: Results are computed for collections of POMs of Manifold$_1$, Manifold$_2$ and Manifold$_3$.

6. Results for Manifold$_1$ + Manifold$_2$ + Manifold$_3$ + Manifold$_4$: Results are computed for collections of POMs of Manifold$_1$, Manifold$_2$, Manifold$_3$ and Manifold$_4$.

## 4 Additional Results

### 4.1 Analyses using Resnets with Different Number of Layers

In this subsection, we give additional results for image classification using Cifar-10 and Imagenet datasets for different networks such as Resnets with 18 and 44 layers (Resnet-18 and Resnet-44), 110-layer Resnets with constant depth (RCD) and stochastic depth (RSD) with data augmentation (DA) and without using data augmentation (w/o DA).

We give classification performance of Resnets with 18 layers (Resnet-18) employed on the Imagenet in Table 8. The results show that performance of CNNs are boosted by employing collections of POMs (denoted by PIO for POMs) using FG-SGD compared to the employment of baseline Euc. We observe that POMs of component manifolds of identical geometry (denoted by POMs of Sp/St/Ob), and their collections (denoted by PIO for POMs of Sp/St/Ob) provide better performance compared to employment of individual component manifolds (denoted by Sp/Ob/St) [9]. For instance, we obtain 28.64%, 28.72% and 27.83% error using PIO for POMs of Sp, Ob and St in Table 8, respectively. However, the error obtained using Sp, Ob and St is 28.71%, 28.83% and 28.02%, respectively. We observe 3.24% boost by construction of an collection of four manifolds

Table 9: Results for Resnet-44 on the Cifar-10 with DA.

| Model | Class. Error(%) |
|---|---|
| Euc. [13] | 7.17 |
| Euc. [9] | 7.16 |
| Euc. † | 7.05 |
| Sp/Ob/St [9] | 6.99/6.89/6.81 |
| Sp/Ob/St † | 6.84/6.87/ 6.73 |
| POMs of Sp/Ob/St | 6.81/6.85/ 6.70 |
| PI for POMs of Sp/Ob/St | 6.82/6.81/ 6.70 |
| PI (Euc.+Sp/Euc.+St/Euc.+Ob) | 6.89/6.84/6.88 |
| PI (Sp+Ob/Sp+St/Ob+St) | 6.75/6.67/6.59 |
| PI (Sp+Ob+St/Sp+Ob+St+Euc.) | 6.31/6.34 |
| PO for POMs of Sp/Ob/St | 6.77/6.83/ 6.65 |
| PO (Euc.+Sp/Euc.+St/Euc.+Ob) | 6.85/6.78/6.90 |
| PO (Sp+Ob/Sp+St/Ob+St) | 6.62/6.59/6.51 |
| PO (Sp+Ob+St/Sp+Ob+St+Euc.) | 6.35/6.22 |
| PIO for POMs of Sp/Ob/St | 6.71/6.73/ 6.61 |
| PIO (Euc.+Sp/Euc.+St/Euc.+Ob) | 6.95/6.77/6.82 |
| PIO (Sp+Ob/Sp+St/Ob+St) | 6.21/6.19/6.25 |
| PIO (Sp+Ob+St/Sp+Ob+St+Euc.) | 5.95/5.92 |

(Sp+Ob+St+Euc.) using the PIO scheme in Table 8 (27.07%). In other words, collection methods boost the performance of large-scale CNNs more for large-scale datasets (e.g. Imagenet) consisting of larger number of samples and classes compared to the performance of smaller CNNs employed on smaller datasets (e.g. Cifar-10). This result can be attributed to enhancement of sets of features learned using multiple constraints.

In addition, we obtain 0.28% and 2.06% boost of the performance by collection of the St with Euc. (6.77% and 28.25% using PIO for Euc.+St, respectively) for the experiments on the Cifar-10 and Imagenet datasets using the PIO scheme in Table 9 and Table 8, respectively. Moreover, we observe that construction of collections using Ob performs better for PI compared to PO. For instance, we observe that PI for POMs of Ob provides 6.81% and 28.75% while PO for POMs of Ob provides 6.83% and 28.81% in Table 9 and Table 8, respectively. We may associate this result with the observation that weights belonging to Ob are used for feature selection and modeling of texture patterns with high performance [28, 35]. However, collections of St and Sp perform better for PO (6.59% and 28.01% in Table 9 and Table 8) compared to PI (6.67% and 28.64% in Table 9 and Table 8) on weights employed on output channels.

It is also observed that PIO performs better than PI and PO in all the experiments. We observe 3.24% boost by construction of an collection of four manifolds (Sp+Ob+St+Euc.) using the PIO scheme in Table 8 (27.07%). In other words, collection methods boost the performance of large-scale CNNs more for large-scale datasets (e.g. Imagenet) consisting of larger number of samples and classes compared to the performance of smaller CNNs employed on smaller datasets (e.g. Cifar-10). This result can be attributed to enhancement of sets of features learned using multiple constraints.

In Table 10, we analyze the performance of larger CNNs consisting of 110 layers on Cifar-100 with and without using DA. We implemented the experiments 10 times and provided the average performance. We observe that sets boost the performance of CNNs that use DA methods more compared to the performance of CNNs without using DA. For instance, PIO of all manifolds (39.35%) outperform baseline (44.65%) by 5.3% without using DA, while those (23.79%) obtained using DA outperform baseline (27.01%) by 3.22% for RCD. Additional results for different CNNs using Imagenet and Cifar-10, and a comparison with vanilla network sets are given in this supplemental material.

Table 10: Classification error (%) for training 110-layer Resnets with constant depth (RCD) and Resnets with stochastic depth (RSD) using the PIO scheme on Cifar-100, with data augmentation (w. DA) and without using DA (w/o DA).

| Model | Cifar-100 w. DA | Cifar-100 w/o DA |
|---|---|---|
| RCD [36] | 27.22 | 44.74 |
| (Euc.) † | 27.01 | 44.65 |
| Sp/Ob/St ([9]) | 26.44/25.99/25.41 | 42.51/42.30/40.11 |
| Sp/Ob/St † | 26.19/25.87/25.39 | 42.13/42.00/39.94 |
| POMs of Sp/Ob/St | 25.93/25.74/25.18 | 42.02/42.88/39.90 |
| PIO (Euc.+Sp/Euc.+St/Euc.+Ob) | 25.57/25.49/25.64 | 41.90/41.37/41.85 |
| PIO (Sp+Ob/Sp+St/Ob+St) | 24.71/24.96/24.76 | 41.49/40.53/40.34 |
| PIO (Sp+Ob+St/Sp+Ob+St+Euc.) | 23.96/23.79 | 39.53/ 39.35 |
| RSD [36] | 24.58 | 37.80 |
| Euc. † | 24.39 | 37.55 |
| Sp/Ob/St [9] | 23.77/23.81/23.16 | 36.90/36.47/35.92 |
| Sp/Ob/St † | 23.69/23.75/23.09 | 36.71/36.38/35.85 |
| POMs of Sp/Ob/St | 23.51/23.60/23.85 | 36.40/36.11/35.53 |
| PIO (Euc.+Sp/Euc.+St/Euc.+Ob) | 23.69/23.25/23.32 | 35.76/35.55/35.81 |
| PIO (Sp+Ob/Sp+St/Ob+St) | 22.84/22.91/22.80 | 35.66/35.01/35.35 |
| PIO (Sp+Ob+St/Sp+Ob+St+Euc.) | 22.19/22.03 | 34.49/34.25 |

Table 11: Classification error (%) for training 110-layer Resnets with constant depth (RCD) and Resnets with stochastic depth (RSD) using the PIO scheme on the Cifar-10, with and without using DA.

| Model | Cifar-10 w. DA | Cifar-10 w/o DA |
|---|---|---|
| RCD [36] | 6.41 | 13.63 |
| (Euc.) † | 6.30 | 13.57 |
| Sp/Ob/St ([9]) | 6.22/6.07/5.93 | 13.11/12.94/12.88 |
| Sp/Ob/St † | 6.05/6.03/5.91 | 12.96/12.85/12.79 |
| POMs of Sp/Ob/St | 6.00/6.01/5.86 | 12.74/12.77/12.74 |
| PIO for POMs of Sp/Ob/St | 5.95/5.91/5.83 | 12.71/12.72/12.69 |
| PIO (Euc.+Sp/Euc.+St/Euc.+Ob) | 6.03/5.99/6.01 | 12.77/12.21/12.92 |
| PIO (Sp+Ob/Sp+St/Ob+St) | 5.97/5.86/5.46 | 11.47/11.65/ 11.51 |
| PIO (Sp+Ob+St/Sp+Ob+St+Euc.) | 5.25/5.17 | 11.29/11.15 |
| RSD [36] | 5.23 | 11.66 |
| Euc. † | 5.17 | 11.40 |
| Sp/Ob/St [9] | 5.20/5.14/4.79 | 10.91/10.93/10.46 |
| Sp/Ob/St † | 5.08/5.11/4.73 | 10.52/10.66/10.33 |
| POMs of Sp/Ob/St | 5.05/5.08/4.69 | 10.41/10.54/10.25 |
| PIO for POMs of Sp/Ob/St | 4.95/5.03/4.62 | 10.37/10.51/10.19 |
| PIO (Euc.+Sp/Euc.+St/Euc.+Ob) | 5.00/5.08/5.14 | 10.74/10.25/10.93 |
| PIO (Sp+Ob/Sp+St/Ob+St) | 4.70/4.58/4.90 | 10.13/10.24/10.06 |
| PIO (Sp+Ob+St/Sp+Ob+St+Euc.) | 4.29/4.31 | 9.52/9.56 |

## 4.2 Comparison with Vanilla Network Ensembles

Our method fundamentally differs from network ensembles. In order to analyze the results for network ensembles of CNNs, we employed an ensemble method [13] by *voting of decisions* of Resnet 44 on Cifar 10. When CNNs trained on individual Euc, Sp, Ob, and St are ensembled using voting, we obtained $7.02\%$ (Euc+Sp+Ob+St) and $6.85\%$ (Sp+Ob+St) errors (see Table 1 for comparison). In our analyses of ensembles (PI, PO and PIO), each POM contains $\frac{N_l}{M}$ weights, where $N_l$ is the number of weights used at the $l^{th}$ layer, and $M$ is the number of POMs. When each CNN in

Table 12: Mean ± standard deviation of classification error (%) are given for results obtained using SENet-Resnet-101, and 110-layer Resnets with constant depth (RCD) on Cifar-100.

| Model Cifar-100 with DA (110 layer RCD) | Error |
|---|---|
| Euc. † | 27.01 ± 0.47 |
| St | 25.39 ± 0.40 |
| POMs of St | 25.18 ± 0.34 |
| PIO (Sp+Ob+St) | 23.96 ± 0.28 |
| PIO (Sp+Ob+St+Euc.) | 23.79 ± 0.15 |
| (Additional results) Cifar-100 with DA (SENet-Resnet-101) | Error |
| Euc. † | 19.93 ± 0.51 |
| PIO (Sp+Ob+St) | 18.96 ± 0.27 |
| PIO (Sp+Ob+St+Euc.) | 18.54 ± 0.16 |

Table 13: Analysis of classification error (%) of state-of-the-art DNNs which employ separable convolutions on Imagenet dataset.

| Model | Classification Error |
|---|---|
| Resnext-50 (Euc. [37]) | 22.2 |
| Resnext-50 (Euc. †) | 22.7 |
| Resnext-50 (Euc.WSS) | 22.3 |
| Resnext-50 (PIO-SOSE) | 21.5 |
| Resnext-50 (PIO-SOSE-WSS) | 21.3 |
| Mobilenetv2 (Euc. [38]) | 28.0 |
| Mobilenetv2 (Euc. †) | 27.9 |
| Mobilenetv2 (Euc.-WSS) | 27.5 |
| Mobilenetv2 (PIO-SOSE) | 26.8 |
| Mobilenetv2 (PIO-SOSE-WSS) | 26.4 |
| DeepRoots (Euc. [39]) | 26.6 |
| DeepRoots (Euc. †) | 27.0 |
| DeepRoots (Euc.-WSS) | 26.6 |
| DeepRoots (PIO-SOSE) | 25.9 |
| DeepRoots (PIO-SOSE-WSS) | 25.5 |

the ensemble was trained using an individual manifold which contains $\frac{1}{4}$ of weights (using $M = 4$ as utilized in our experiments), then we obtained 11.02% (Euc), 7.76% (Sp), 7.30% (Ob), 7.18% (St), 9.44% (Euc+Sp+Ob+St) and 7.05% (Sp+Ob+St) errors. Thus, our proposed methods outperform ensembles constructed by voting.

## 4.3 Analyses for Larger DNNs with Large Scale Image Datasets

We give the results for Cifar-100 obtained using data augmentation denoted by with DA in Table 12.Cifar-100 dataset consist of $5 \times 10^4$ training and $10^4$ test images belonging to 100 classes.

In Table 12, we provide results using the state-of-the-art Squeeze-and-Excitation (SE) blocks [34] implemented for Resnets with 110 layers (Resnet-110) on Cifar-100. We run the experiments 3 times and provide the average performance.

In the second set of experiments, we perform separable convolution operations using the proposed weight splitting scheme. We compare the results using various popular separable convolution schemes, such as depth-wise and channel-wise convolution implemented using state-of-the-art DNNs such as ResNext with 50 layers (ResNext-50) [37], MobileNet v2 with 21 layers (Mobilenet) [38] and 50 layer Resnets with hierarchical filtering using 4 roots (DeepRoots) [39]. The results obtained using PIO with (Sp+Ob+St+Euc.) with the separable convolution scheme proposed in the corresponding related work are denoted by PIO-SOSE. The results obtaied using PIO with (Sp+Ob+St+Euc.) with our proposed WSS are denoted by PIO-SOSE-WSS.

Table 14: Mean ± standard deviation of classification error (%) are given for results obtained using Resnet-50/101, SENet-Resnet-50/101, and 110-layer Resnets with constant depth (RCD) on Imagenet.

| Model | Imagenet(Resnet-50) | Imagenet(SENet-Resnet-50) |
|---|---|---|
| Euc. | 24.73 ± 0.32 | 23.31± 0.55 |
| St | 23.77 ± 0.27 | 23.09 ± 0.41 |
| POMs of St | 23.61 ± 0.22 | 22.97 ± 0.29 |
| PIO (Sp+Ob+St) | 23.04 ± 0.10 | 22.67 ± 0.15 |
| PIO (Sp+Ob+St+Euc.) | 22.89 ± 0.08 | 22.53 ± 0.11 |
| (Additional results) | Imagenet(Resnet-101) | Imagenet(SENet-Resnet-101) |
| Euc. | 23.15 ± 0.09 | 22.38 ± 0.30 |
| PIO (Sp+Ob+St) | 22.83 ± 0.06 | 21.93 ± 0.12 |
| PIO (Sp+Ob+St+Euc.) | 22.75 ± 0.02 | 21.76 ± 0.09 |

Table 14 shows results using the state-of-the-art Squeeze-and-Excitation (SE) blocks [34] implemented for Resnets with 50 layers (Resnet-50) on Imagenet. We run the experiments 3 times and provide the average performance. We first observe that PIO boosts the performance of baseline Euc. (24.73%) by 1.84% if sets of weights are employed using Euc, Sp, Ob and St (22.89%). We note that the sets computed for Resnet-50 outperform Resnets with 101 layers (23.15) by 0.26%. SE blocks aim to aggregate channel-wise descriptive statistics (i.e. mean of convolution outputs) of local descriptors of images to feature maps for each channel. In FG-SGD, we use standard deviation (std) of features extracted from each batch and size of receptive fields of units while defining and updating weight manifolds (see Section 3.3 in supp. mat.). Unlike SE blocks, FG-SGD computes statistical and geometric properties for different sets of input and output channels, and used to update weights by FG-SGD. This property helps FG-SGD to further boost the performance. For instance, we observe that collections of manifolds (23.04% and 22.89% error) outperform SENet-Resnet-50 (23.31% error). Although FG-SGD estimates standard deviation using moving averages as utilized in batch normalization [40], SE blocks estimates the statistics using small networks. Therefore, we conjecture that they provide complementary descriptive statistics (mean and std). The experimental results justify this claim such that sets implemented in SENet-Resnet-50 further boost the performance by providing 22.53% error.

## Footnotes

[1] see Theorem 24 in [8] .

[2]For the example of training using the Cifar-100 dataset given above.

[3]For different implementations of QR decomposition on GPUs, we observed $3 < \mathfrak{a} < 6$.

[4]We observed that for Intel Xeon E5-1650 v3, and obtained improvement of running time by approximately $f(D_l) < 5$ for E5-2697 v2 since using larger number of CPU cores.