[Reviews · NeurIPS 2019]

Reviewer 1



The work is novel and significant. It applies previously established generalization bounds in the form of upper bounds on norms of DNN weights towards providing performance guarantees by instead imposing these bounds as constraints on the parameters. This is an interesting development studying the geometry of parameters W of deep networks, and is a promising direction to pursue as demonstrated by improved performance in experimental results in practice. Authors describe challenges of handling multiple constraints expressed as products of manifolds, as they can be different from the geometry imposed by individual component manifolds. Further they provide the algorithm FG-SGD as a way to modify SGD such that the POM constraints are satisfied. The modification involves projection operation onto the tangent space of the manifold. Rescaling operation so that the upper bound RHS is 1. Clarity can be improved by avoiding run-on sections and breaking up text into meaningful subsections. clearly stating the purpose of each section, the motivations, findings and conclusions. ### After rebuttal, review discussion: Suggestions to improve clarity - Reduce introduction to no more than 1 page - List contributions as done in rebuttal succinctly - Currently, there is some redundancy under - Boundary names of lists - Training DNNs and turn another bulleted list in pages 2 and 3 Please consolidates - Have a separate background/related work section to motivate work - Have a separate notation section, introducing any notation used throughout paper. It is difficult to parse when new notation is introduced right before it is used or not at all - Subsections in section 3 e.g., -- "In order to ..." should be a new section -- The 2 results should be listed as sub-headings -- How these results are incorporated into algorithm should be listed immediately after Section 4: - Compress table 1 & 2 captions - Notation to be described separately and not merged in bullets - Derivation of Lines 5,6,7 should be explained in greater detail

Reviewer 2



This is probably the densest paper that this reviewer has seen in a long time. The material has been squashed, almost certainly beyond the point of breaking, to fit within the 8-page limit. This is exemplified by the fact that the supplementary document is 19 pages in length. It would seem to make more sense to present this in a longer format - for example a journal article which would allow for a more clear explanation of the ideas. Or alternatively the paper could (potentially) be split up into separate papers. As such the clarity of the presented work is poor which makes analysis of the quality and significance difficult to assertion.

Reviewer 3



-------post-rebuttal comments The authors' response addresses all of my minor improvement suggestions. So I am still very positive about this paper. -------pre-rebuttal comments I vote for accept this paper due to: 1) the authors proposed a reasonable hypothesis that impose multiple constraints on DNN weights to bound its norm can get empirical generalization errors closer the theoretical bounds, and also improve performance. 2) they designed a novel algorithm FG-SGD to achieve the goal, while they addressed a lot of technique difficult problems. 3) They conducted solid experiments to support the superior of the proposed method. See details breakdown comments into review rubrics below. 1. Originality - 9 The proposed FG-SGD algorithm is novel. There are also a lot of non-trivial technique contribution of the work to make training with complicated constraints over POMs feasible, in which it cannot be done by simply applying existing techniques. 2. Quality - 9 Although I didn't follow all the mathematical details carefully in the paper, the overall story is sound and the mathematical analysis seems to be very solid. The author also conducted extensive experiments over widely used ResNet50/101, ResNext, MobileNet and DeepRoot architecture on multiple datasets Cifar-10, Cifar100 and Imagenet to support the claim of the paper, and the superior of the proposed method. 3. Clarity - 7 This paper is very mathematical dense and heavy so that it takes effort to follow, even though the authors spent a lot of effort to provide high level ideas in the main paper to help readers understand it. It will also be great if the author can release the code to help other people to understand and implement the algorithms, and reproduce the results. 4. Significance - 8 This works seems to be applicable to all the existing NN learning problems. The only question is whether the extra training time caused by the complexity of the proposed algorithm pay off. If the authors can integrate this algorithm into standard ML library like TensorFlow or PyTorch, I am sure it will help people to use it.

[Author Response · NeurIPS 2019]

We thank reviewers for their comments and suggestions. Please find below our point-to-point response.

**R1, R2, R3; Contextualization, decompression and a concise summary of the present work:** We agree with reviewers on splitting the sections into subsections to articulate and present the following crucial ideas and results more clearly. We will update this manuscript accordingly. In order to provide more detailed theoretical and experimental analyses and results, we have been preparing an extended version of the work (e.g. as a technical report/journal paper) with a toolbox/API supporting PyTorch, Tensorflow and MXNet. In this submitted version of the work, we introduce an overview of a unified mathematical and algorithmic framework which can be used to train DNNs employing different constraints on weights, with concrete generalization and convergence properties, and improving accuracy of baselines.

**Definition of the proposed major problem:** Generalization errors of DNNs are bounded by functions of various norms of weights of the DNNs. Our major goal is improvement of their generalization error by training DNNs according to these norms, under a unified algorithmic framework with precise generalization and convergence properties.
**Proposed solutions for the major problem:** We propose to (i) learn bounds of norms of weights, and (ii) optimize the weights with bounded norms for training of DNNs with better generalization error/accuracy in theory and practice.
**Subproblems:** The problem (i) is posed as estimation and learning of bounds of norms using geometry of spaces of feature representations and weights, and statistical properties of data and features, during training. However, weights with varying bounded norms reside on different manifolds, and their geometric properties (i.e. metrics and curvatures) change as bounds are updated while resolving (i) during training. Thus, the problem (ii) is posed as joint optimization of multiple fine-grained weights with different norms residing on products of the corresponding manifolds endowed with dynamically changing geometry with guarantee of convergence to local and global minima (Section 2 and 3).
**Proposed solutions for the subproblems and results:** To solve (i), we propose a two-stage re-normalization method by first bounding norms of weights to 1.0, and then learning the upper bounds of norms according to dimension of feature spaces and receptive fields determined by weights, and standard deviation of data and features. We also provide bounds and values of norms as functions of these geometric and statistical properties in Table 1 and 2. To solve (ii), we propose the FG-SGD in Section 4, and provide theoretical and experimental results in Sections 4, 5 and supp. mat.

**R1, R2; Employment of shallow methods for optimization on product manifolds in DNNs using SGD, and related work:** We consider [17,18] as two related works which optimize weights on particular *static product manifolds* to train shallow models. When we apply these methods for optimization on product of two or more *dynamic manifolds* in DNNs using SGD, Hessian of geodesic of the product manifold may not be bounded. In this case, we observe early divergence due to exploding or vanishing gradients. To this end, we first analyze relationship between geometry of product and component manifolds (i.e. metrics and geodesics) in Section 3. Then, we employ these results to bound gradients and Hessian on the product manifolds using those of component manifolds in Section 4, while developing the FG-SGD. Our proposed approach can be used to extend optimization methods proposed in recent related works, such as those proposed by Mishra et al., Sato et al., Huang et al., to apply their methods with dynamic product manifolds in DNNs. We will provide this discussion in the final version of the paper with the additional aforementioned related work.

**R1; A sketch of proof idea, and equations (4), (5) and (6):** In this work, we develop our algorithms by employing theoretical results using mathematical methods in their implementation. More precisely, the constraints and lemmas/theorems used to prove convergence theorems (Theorem 2 and Corollary 1) are realized and implemented in the algorithms. In order to introduce and explain this approach, we provided an overview of properties of geometry of manifolds used to prove convergence theorems in Section 3. Mathematical assumptions and steps of the proofs of convergence theorems are realized and implemented in the steps (Line 5, 6, 7) of the Algorithm 1. Remark 1 (Lemma 1) given in Section 3 was used to prove Theorem 1 which was used to prove convergence theorems. The equation (6) of Theorem 1 was used to compute functions given in the equations (3) and (5), and the equation (4) is a constraint used to compute learning rate in (3) at Line 6 of the Algorithm 1. These functions were used to prove the convergence theorems. A method used to compute a particular step size in (7) was proposed as a realization of Corollary 2. Therefore, we agree that the overall proof sketch was distributed in different sections of the paper. We consider providing an overview and a graphical sketch of proof of the theorems in Section 3 following suggestion of R1.

**R3; Running time:** Training time of DNNs for Euc, Sp, and Ob are similar. Training time for the St is affected by running time of matrix decomposition methods used by some projections, depending on numerical library and computer systems. Therefore, we provided a theoretical analysis of their computational complexity. When we apply approximation methods such as singular value bounding or power iteration for projections, then running times for the St approach to those of the other manifolds. For instance, for the experiments given in Table 3, the best running times (images/second) on a Tesla P100 are approximately: 200 (Euc, Sp, Ob), 180 (St), 185 (Sp+Ob+St), 190 (Sp+Ob+St+Euc).

**R3; Results on NMT:** Thank you for the notification, and we will fix the statement. We removed results obtained for NMT tasks to reduce complexity of presentation of the work and focus on image classification tasks. As an ablation study and a proof of concept, we obtained the following BLEU scores using a transformer network (Vasvani et al., NIPS'17) for English to German translation on the WMT newstest2014: Baseline (27.1), Sp+Ob+St+Euc (28.3).

[Meta-Review · NeurIPS 2019]

The reviewers were generally impressed by the quality of technical contributions, but had concerns about the clarity of the work. In particular, there was a shared sense that the paper was very densely written and difficult to understand. The authors promise many revisions, which reviewers are taking in good faith. Reviewer 1 has left very detailed comments about how to revise the paper to improve clarity. Please take these very seriously and undertake to make revisions with these points in mind for a final version.